

# Impacts of solar-absorbing aerosol layers on the transition of stratocumulus to trade cumulus clouds

Xiaoli Zhou[1], Andrew S. Ackerman[2], Ann M. Fridlind[2], Robert Wood[3] and

Pavlos Kollias[4,5]

1. Department of Atmospheric and Oceanic Sciences, McGill University, Montreal,

Quebec, CA

2. NASA Goddard Institute for Space Studies, New York, New York, USA

3. University of Washington, Seattle, Washington, USA

4. School of Marine and Atmospheric Sciences, Stony Brook University, Stony Brook, New

York, USA

5. Department of Environmental and Climate Sciences, Brookhaven National Laboratory,

Upton, New York, USA

*Correspondence to*: Xiaoli Zhou (xiaoli.zhou@mail.mcgill.ca)





## Abstract

The effects of an initially overlying layer of solar-absorbing aerosol on the transition of stratocumulus to trade cumulus clouds are examined using large-eddy simulations. The transition of lightly drizzling cloud is generally hastened, resulting mainly from increased

cloud droplet number concentration ($N_c$) induced by entrained aerosol. The increased $N_c$ slows sedimentation of cloud droplets and shortens their relaxation time for diffusional growth, both of which accelerate entrainment of overlying air and thereby stratocumulus breakup. However, the decrease in albedo from cloud breakup is more than offset by redistributing cloud water over a greater number of droplets, such that the diurnal-average

shortwave forcing at the top of atmosphere is negative. The negative radiative forcing is enhanced by sizable longwave contributions, which result from the greater cloud breakup and a reduced boundary layer height associated with aerosol heating. A perturbation of moisture instead of aerosol aloft leads to greater liquid water path and a more gradual transition. Adding absorbing aerosol to that atmosphere results in substantial reductions

in LWP and cloud cover that lead to positive shortwave and negative longwave forcings on average canceling each other. Only for heavily drizzling clouds is the breakup delayed, as inhibition of precipitation overcomes cloud water loss from enhanced entrainment. Considering these simulations as an imperfect proxy for biomass burning plumes influencing Namibian stratocumulus, we expect regional indirect plus semi-direct

forcings to be substantially negative to negligible at the top of atmosphere, with its magnitude sensitive to background and perturbation properties.



## 1. Introduction

25       Aerosols affect the earth's radiation budget in at least three ways. First, they directly absorb and scatter solar radiation. Second, they affect radiative fluxes indirectly through their role as cloud condensation nuclei, influencing cloud microphysics and thereby affecting cloud albedo and cloud cover. Third, solar-absorbing aerosols can alter atmospheric heating rates and stability, leading to rapid adjustments in cloud properties;

the resulting impact on radiative fluxes is referred to as the semi-direct effect (Hansen et al., 1997).

      Aerosols have been identified as contributing the greatest uncertainty to anthropogenic climate forcing (Forster et al. 2007). For instance, with regard to semi-direct forcings, some general circulation model (GCM) studies (e.g., Hansen et al., 1997;

Lohmann and Feichter, 2001; Jacobson, 2002; Cook and Highwood, 2004) have found a net decrease in low-level cloud cover, which corresponds to a positive radiative forcing at the top of the atmosphere (TOA) that tends to warm the climate system, while others (e.g., Menon et al., 2002, Penner and Zhang, 2003; Sakaeda et al, 2011) have found the opposite, in which the cloud water increases and the radiative forcing depends crucially

on the height of the absorbing aerosol. To better constrain radiative forcing in climate models, a comprehensive understanding of regional cloud-aerosol interactions and the corresponding radiative forcings is of value.

      Here we focus on warm (liquid-phase) clouds in the planetary boundary layer (PBL). Process-level understanding of the physical mechanisms underlying indirect and

semi-direct aerosol radiative forcings has been largely advanced through studies with large-eddy simulation (LES) models and in situ observations. Regarding aerosol indirect





forcing, with all else equal (particularly cloud cover and liquid water path), increased cloud droplet number concentration ($N_c$) resulting from increased aerosol concentration ($N_a$) increases cloud optical thickness and thus albedo, thereby exerting a negative

radiative forcing at TOA (Twomey 1974, 1991). For precipitating clouds, increasing $N_c$ can reduce precipitation and thereby enhance liquid water path (LWP) and cloud cover (e.g., Albrecht, 1989; Ackerman et al., 1993; Pincus and Baker, 1994; Hindman et al., 1994). However, for clouds with little precipitation, modeling studies indicate that increased $N_c$ tends to reduce LWP and cloud cover by increasing PBL entrainment

(Ackerman et al., 2004; Wood et al., 2007; Ackerman et al., 2009), which can dry the PBL and reduce LWP when the overlying air is sufficiently dry (Randall, 1984). Such a tendency is consistent with satellite observations of LWP reduction in ship tracks, on average (Coakley and Walsh, 2002). At least three microphysical mechanisms have been found to play a role in the entrainment increase. First, in what we shall refer to as the

"sedimentation effect", increased $N_c$ leads to smaller droplets that fall more slowly, which increases the amount of cloud water available for evaporative cooling during entrainment events, thereby strengthening entrainment (Bretherton et al., 2007). Second, in what we shall refer to as the "evaporation effect", smaller droplets increase the total surface area of cloud droplets, accelerating evaporation and driving stronger entrainment (Xue et al.,

2008). Third, increased $N_c$ also suppresses drizzle, enhancing convective intensity and entrainment (e.g., Stevens et al. 1998, Wood et al. 2007). Under dry overlying air, all three effects tend to reduce cloud cover and LWP, leading to a positive radiative forcing. However, if the entrained air is sufficiently moist, entrainment can be expected to increase LWP (Randall, 1984).



Aerosol semi-direct effects have been studied by Ackerman et al. (2000) in the context of trade cumulus under a sharp inversion, in which absorbing aerosol within the boundary layer increases solar heating in a manner that stabilizes the PBL, reducing the moisture supply from the surface and the amount of cloudiness, leading to a positive radiative forcing at TOA. More directly in such a scenario the relative humidity of the

PBL is reduced by enhanced solar heating, reducing cloudiness as originally found in global model simulations by Hansen et al. (1997). In contrast, Johnson et al. (2004) conducted large-eddy simulations of marine stratocumulus and found that an absorbing aerosol immediately above the PBL (and not entrained) strengthens the inversion, reducing entrainment and thereby increasing cloud cover, leading to a negative radiative

forcing, while they found the opposite (positive radiative forcing) for aerosol heating within the PBL. That study was motivated at least in part by measurements of absorbing aerosol from biomass burning advected from Africa over Namibian stratocumulus, where biomass burning aerosol plumes may also be well separated from the PBL (Keil and Haywood, 2003, Haywood et al., 2003b), a factor that has been found to be critical to

absorbing aerosol effects on cloud fraction (Feingold et al., 2005).

Further complexity arises when considering the possibility that absorbing aerosol can act as cloud condensation nuclei (CCN) and thereby increase $N_c$, which was neglected in the early studies of Johnson et al. (2004) and Feingold et al. (2005) and only represented quite crudely by Ackerman et al. (2000), who simply imposed a sequence of

uniform $N_c$ values in their simulations. Here we will consider both roles of absorbing aerosol.



By considering two trade cumulus regimes, one transitional case with a sharp inversion (ATEX) and a more downstream case with greatly reduced cloud cover (BOMEX), Johnson (2005) found the semi-direct aerosol forcing to depend strongly on

the cloud regime, with the magnitude of the forcing increasing with (unperturbed) cloud cover. This regime dependence is relevant to the stratocumulus-to-cumulus transition (SCT), a climatological feature downstream of subtropical marine stratocumulus (Klein and Hartmann, 1993; Sandu et al., 2010; Zhou et al., 2015). The SCT has been found in modeling studies to be driven by equatorward advection over increasing sea surface

temperatures (SST), which increases surface latent heat fluxes, enhancing buoyancy fluxes in the cloud layer and hence entrainment. The PBL deepening from progressive entrainment inhibits the ability of circulations forced at cloud top to maintain a well-mixed boundary layer, reducing the surface moisture supply and eventually drying out the stratocumulus clouds (Bretherton and Wyant, 1997; Wyant et al., 1997). A recent

observational study has found that the time scale of the SCT over the eastern Pacific can depart considerably from that in an idealized model framework driven only by increasing SST (Zhou et al., 2015), suggesting that other factors, such as meteorological variability, might play important roles in the time scale of SCT. Yamaguchi et al. (2015) (hereafter Y15) investigated the impact of overlying absorbing aerosol and associated enhanced

moisture on the SCT and found that entrained absorbing aerosol in general delays the SCT with a net negative change in TOA shortwave (SW) cloud radiative forcing (CRF).

It has been documented in recent observational studies near northern Namibia and remote St. Helena Island in the South Atlantic Ocean that the sampled absorbing aerosol is often accompanied by enhanced humidity, with an average moisture perturbation of



~1 g kg$^{-1}$ relative to the underlying air (Haywood et al., 2003b; Adebiyi et al. 2015). This humidity is associated with the outflow from the deep, continental boundary layer. The enhanced humidity induces additional radiative heating, which can regulate cloud processes by reducing cloud-top longwave (LW) cooling (Adebiyi et al. 2015; hereafter A15) and by simply reducing the dryness of air entrained into the PBL. Y15 located a

stationary moist layer above the PBL and found that the additional moisture itself enhances cloud breakup during the SCT, although they acknowledge that their perturbation of ~3 g kg$^{-1}$ likely represents an upper limit compared with A15.

Here we perform an expanded investigation of the impact of absorbing aerosol and moisture on the SCT. Because Y15 was published during the course of this work, our

simulation setups are similar but not identical, and we highlight similarities and differences below. Like Y15, we adopt the Sandu and Stevens (2011) SCT case study, with some modifications. Here we separate the responses to aerosol heating above and within the PBL and on microphysical processes. We consider the impacts on lightly and heavily drizzling stratocumulus decks. We also assess the impacts of additional overlying

moisture on the SCT and how it influences the effects of absorbing aerosol. The radiative forcings in our study consider not only changes in SW but also LW fluxes. Our results differ from Y15 in that initially overlying plumes of absorbing aerosol lead to positive changes in SW CRF at TOA, and the aerosol and moisture perturbations never delay the SCT in our simulations (unless we omit well-established physical processes).

The remainder of this manuscript is organized as follows. Section 2 documents the model setup and case description. Section 3 presents analysis of the microphysical and heating effects of absorbing aerosol during the transition of lightly drizzling



stratocumulus. In sect. 4, we investigate the impact of additional moisture in the aerosol

layer, and the influence of the initial altitude of the moist aerosol layer. The impacts of an

absorbing aerosol on the SCT of heavily drizzling stratocumulus are discussed in sect. 5.

In sect. 6 we discuss and summarize our findings.

### 2.  Model setup and simulated cases

The Distributed Hydrodynamic Aerosol and Radiative Modeling Application

(DHARMA) (Ackerman et al., 2004 and references therein) simulations here are based

on the "reference case" 3-day Lagrangian SCT setup of Sandu and Stevens (2011). The

basis for the case is a composite of the large-scale conditions encountered along

trajectories over the northeast Pacific from June to August of 2006 and 2007. An

intercomparison of six different LES models shows that DHARMA results are consistent

with others in representing the SCT (de Roode et al., 2016), although differences between

models do exist, as discussed further below. Unlike Sandu and Stevens (2011) and Y15,

here we begin simulations at midnight local time (when turbulent mixing is vigorous, to

accelerate spin-up) rather than 10:00 local time.

The DHARMA domain size is 10.8 km x 10.8 km x 3.2 km and horizontal

resolution is set to $\Delta x = \Delta y = 75$ m. Vertically 240 levels are distributed between 0 and

3200 m, with variable vertical resolution ranging from 30 m near the surface to 10 m near

the inversion and up to 60 m near the model top; before using this grid with twice as

coarse of a grid as in de Roode et al. (2016), we confirmed that the DHARMA results

were not sensitive to the difference. The microphysics scheme is an adaptation of the

two-moment scheme of Morrison et al. (2005) with prognostic saturation excess



following Morrison and Grabowski (2008) and assuming the shape factor of the cloud droplet size distribution to be 10.3 (equivalent to relative dispersion of 0.3) following Geoffroy et al. (2010). Activation of aerosol follows Abdul-Razzak and Ghan (2000) using supersaturation computed after the condensational adjustment of Eq. A10 in

Morrison and Grabowski (2008). The aerosol are semi-prognostic: prognostic in that the number concentration of unactivated plus activated aerosol for each species is prognostic (advected), but there is no evolution of the size and breadth of the underlying aerosol size distribution for each species nor are there sources or sinks of aerosol number, and thus the scheme is diagnostic in the sense that total particle number concentration is conserved.

170        Two species of aerosol are prescribed: ammonium sulfate and a solar-absorbing aerosol; both aerosol types act as CCN and interact with the radiation before and after activation. The baseline case is an ensemble of three simulations with different pseudo-random seeds for the initial temperature perturbation field in the PBL, and includes only ammonium sulfate aerosol, which are uniformly distributed in the vertical with

$N_{a,\,sulfate} = 150$ mg$^{-1}$ (without a vertical gradient the aerosol scheme is completely diagnostic). Further simulations are conducted that incorporate an absorbing aerosol profile initialized to increase linearly from zero below 1250 m altitude up to $N_{a,\,absorb} = 5000$ mg$^{-1}$ at 1300 m, maintain a uniform value up to 2800 m, then decrease to zero at 2850 m and above. Log-normal size distributions are specified for the sulfate and

absorbing aerosol, with geometric mean radii of 0.05 μm and 0.12 μm and geometric standard deviations of 1.2 and 1.3, respectively. The hygroscopicity parameter κ (Petters and Kreidenweis, 2007) is set to 0.55 for ammonium sulfate and 0.2 for the absorbing aerosol. The size distribution for the absorbing aerosol is based on the measurements of



Haywood et al. (2003b) and the hygroscopicity (for aged biomass burning aerosol) from

those of Englehart et al. (2012). The absorbing aerosol optical properties follow the

approach of Ackerman et al. (2000) but here a soot core radius of 0.04 μm is specified,

resulting in a single scattering albedo (SSA) of 0.88 at wavelength 0.55 μm. The

extinction coefficient within the absorbing aerosol layer is about 0.16 km$^{-1}$ at 0.55 μm,

consistent with the measurements reported by Haywood et al. (2003a). The absorbing

aerosol induces a heating rate of ~2.6 K d$^{-1}$ at noon and a diurnal-average heating rate

~1.2 K d$^{-1}$, consistent with observations exploited by Johnson et al. (2004) and Ackerman

et al. (2000). The initial absorbing aerosol layer physical thickness of 1.5 km is loosely

based on observations over the southeast Atlantic by Chand et al. (2009), Haywood et al.

(2003b), and Labonne et al. (2007), who report characteristic layer thickness over the

Atlantic of 1 to 2 km. Sensitivities of the results to the assumed SSA of the absorbing

aerosol and to their initial number concentration are briefly discussed.

To examine variations in bulk properties of the overlying aerosol layer, a further

simulation is performed with the initial location 400 m higher, in which the model

domain is extended to 3.5 km and the column of overlying water vapor and ozone used

for radiative fluxes adjusted accordingly. An additional baseline case with a 3.5-km deep

grid was run for computing differences. Two other simulations consider a moist

perturbation of 1 g kg$^{-1}$ based on observations at St. Helena Island of outflow from the

continental boundary layer (A15), scaled to the initial height of $N_{a, absorb}$ with and without

absorbing aerosol. Finally, the impact of overlying absorbing aerosol on heavily

precipitating stratocumulus is examined by reducing $N_{a, sulfate}$ to 25 mg$^{-1}$. To isolate the

microphysical effects of the overlying aerosol, a group of simulations is performed where





the interaction of the absorbing aerosol with radiation is omitted. The aforementioned sedimentation and evaporation effects are examined by additional simulations that exclude cloud droplet sedimentation and that fix the cloud droplet relaxation time scale

(instead of computing it per Equation A5 of Morrison and Grabowski, 2008). Semi-direct aerosol effects are dissected through simulations that restrict aerosol heating to the free troposphere (FT) or the PBL.

Radiative forcings are computed from hourly time slices, which yield daily averages that differ negligibly from those using radiative fluxes updated every minute.

We compute aerosol forcings following Ghan (2013), in which total forcing from a perturbation is calculated as the change in net downward radiative flux at TOA relative to the baseline: $\Delta F = F(\text{perturbed}) - F(\text{baseline})$. The sum of the indirect and semi-direct forcings from the absorbing aerosol is computed similarly but with the absorbing aerosol omitted when calculating $F(\text{perturbed})$. The direct aerosol forcing is then derived by

subtracting the sum of indirect and semi-direct forcings from the total forcing.

For the sake of comparison with Y15, in one instance we also compute cloud radiative forcing as the difference of net downward radiative fluxes at TOA with and without cloud: $F(\text{all sky}) - F(\text{clear sky})$. The difference between $\Delta F$ and the aerosol-induced change in cloud radiative forcing is the direct aerosol forcing for clear sky:

$\Delta CRF = \Delta F - \Delta F(\text{clear sky})$. The enhancement of aerosol absorption associated with SW reflection by an underlying cloud layer, which tends toward a positive forcing (e.g., Chand et al., 2009) and is implicitly included in $\Delta F$, is offset in $\Delta CRF$ by the subtraction of a direct forcing that tends more negative here, because the ocean surface is less




reflective than the cloud layer. Subtraction of a negative direct forcing thereby yields a

ΔCRF that tends to be more positive than total forcing Δ$F$.

In all forcing calculations for this study, net LW fluxes at TOA are scaled from

net LW fluxes at the top of the model domain using $F_{\text{TOA}} = 2.627F_{3.2\text{km}} + 0.0054F_{3.2\text{km}}^2$

for the 3.2-km deep grid, and using $F_{\text{TOA}} = 2.469F_{3.5\text{km}} + 0.0046F_{3.5\text{km}}^2$ for the 3.5-km

deep grid. These correlations were derived from the baseline case run on a 40-km deep

grid, with root mean square (RMS) errors of 0.3 and 0.2 W m$^{-2}$ on the shallower grids,

with biases of less than 0.001 W m$^{-2}$. No TOA corrections for SW fluxes are made

because the radiative transfer scheme (Toon et al., 1989) provides accurate TOA fluxes

by treating Rayleigh scattering in the overlying atmosphere.

**3. Impacts on lightly drizzling SCT**

**3.1. Overview of SCT with and without absorbing aerosol layer**

Figs. 1 and 2 illustrate the transition from a compact stratocumulus layer to more

broken fields of cumulus as a response to increasing SST for the lightly drizzling baseline

case ($N_{\text{a, sulfate}}$ = 150 mg$^{-1}$, $N_c$~100 cm$^{-3}$). The PBL depth in general increases with SST

and reaches 2 km at the end of day 3 (Fig. 1a). The thinning of the stratocumulus is

observed in the afternoon of day 1 as solar heating offsets some of the LW cooling that

drives PBL mixing, when vertical wind variance profiles show bimodal structure with a

local minimum near cloud base (~12 h in Fig. 1b). Convection revitalizes after sunset and

deepens the stratocumulus. Starting around sunrise of day 2 (~30 h), the PBL becomes

continuously stratified, with a persistent cumulus layer developing under the

stratocumulus (Fig. 1a). This stratification reduces the subsequent nocturnal recovery,



and leads to further reduction in LWP (Fig. 2b) and cloudiness (Fig. 2c) after sunrise on day 3. Following Sandu and Stevens (2011) by defining the SCT as the time at which cloud cover (the fraction of columns with LWP > 10 g m$^{-2}$) first decreases to half of its

initial value, the transition in the baseline case is at ~62 h.

When incorporating an overlying absorbing aerosol layer, the clouds and PBL evolve in a notably different way with an evident radiative impact (Figs. 2 and 3; Table 1). $N_c$ increases gradually after the bottom of the ramp of subsiding aerosol contacts the deepening PBL at ~15 h (Fig. 2a). The full strength of the aerosol layer reaches the PBL

at ~20 h (Fig. 2d). Before the subsiding aerosol layer contacts the deepening PBL, absorption of SW radiation in the aerosol layer dominates the radiative impact and reduces the diurnal-average upwelling SW radiative fluxes at TOA by ~7 W m$^{-2}$ on day 1 (Fig. 2f, Table 1). This SW absorption by the aerosol layer decreases with time when the cloud field is more broken, since less upwelling SW radiation is reflected back into the

layer (cf. Chand et al., 2009) and when it is mixed below cloud, where less SW radiation reaches the absorbing aerosol. On day 3, SW absorption is overcome by scattering, resulting in a negative direct forcing (Table 1).

As the absorbing layer approaches the PBL, the inversion strengthens (Fig. 2h), which would tend to slow entrainment. However, as the layer makes contact with the

clouds, the entrained aerosol activate cloud droplets and lead to a pronounced increase of $N_c$, which is ultimately increased by a factor of ~10 over the baseline to ~1000 cm$^{-3}$ (Fig. 2a). The increased $N_c$ acts to accelerate entrainment through the sedimentation and evaporation effects, and opposes but does not overcome the opposing tendency from the strengthening of the inversion (Figs. 2d and 2e). The entrainment of warmer air with less



RH leads to a reduction of LWP (Fig. 2b) and cloud cover (Fig. 2c), hastening and

enhancing the SCT on day 2 (Fig. 2c). This SCT acceleration is opposite to Y15 who

found that entrained absorbing aerosol delays the SCT and leads to overcast conditions

during the second half of 72-h simulations. As a result of substantially reduced LWP,

here the overlying absorbing aerosol case yields a positive change in TOA SW CRF

relative to the baseline during the 3-day simulation (Table 2). The daytime average SW

$\Delta$CRF after the soot contacts the PBL is 9.3 W m$^{-2}$, opposite in sign to that of Y15.

Meanwhile, the negative LW contributions to $\Delta$CRF are enhanced during the transition,

and overcome the positive SW $\Delta$CRF on day 3. As explained further below, such LW

contributions result from microphysical and heating effects. While such LW forcings are

often ignored when considering aerosol impacts on low-lying clouds, much of the

subtropical and tropical atmosphere is not particularly moist, with column water vapor of

less than 30 mm (cf. Lindstrot et al. 2014) as it is here (initial and final values

respectively about 25 and 30 mm), allowing changes in low-level clouds to impact LW

fluxes at TOA.


### 3.2 Microphysical effects

The microphysical effects of the subsiding aerosol are isolated by omitting aerosol

heating and comparing to the same baseline (Fig. 4). The substantial increase of $N_c$ as a

result of the entrained aerosol is seen to largely explain overall reductions of both LWP

and cloud cover relative to the baseline simulation, leading to a hastened SCT. Such

disparity in LWP and cloud cover with and without entrained aerosol is reduced when

either the sedimentation effect is excluded (by omitting cloud droplet sedimentation from



both simulations) or when the evaporation effect is excluded (by fixing the cloud droplet

diffusional growth relaxation time in both simulations). When both effects are excluded,

simulations with and without entraining aerosol exhibit negligible differences in LWP

and a reversed difference in cloud cover. Thus, the hastened SCT from absorbing aerosol

in DHARMA simulations can be attributed primarily to the microphysical effects of

increased $N_c$, specifically via sedimentation and evaporation effects.

With the semi-direct effect now excluded by omitting aerosol absorption, the

indirect forcing is isolated (Table 3). Despite the substantial reduction in cloud cover, the

entrained aerosol results in only a modest positive aerosol indirect forcing on day 2 and a

negative forcing on day 3 (Table 3). The negative forcing is driven by a negative LW

forcing, as a result of more broken clouds and emission from a warmer SST, and by a

significant Twomey effect, which does not fully offset the opposed, comparable SW

forcing induced by the sedimentation and evaporation effects (Table 4).

### 3.3 Semi-direct effect

Next we isolate the semi-direct effect of aerosol heating by considering aerosol

absorption in the FT, PBL and throughout the atmosphere and comparing to the

preceding case that only included microphysical effects of the entrained aerosol layer. As

seen in Fig. 5, aerosol heating in the FT substantially strengthens the PBL inversion as

the aerosol layer approaches the PBL (Fig. 5e), enhancing LWP and cloud cover (Figs. 5b

and 5c) by inhibiting entrainment (Fig. 5d). The increase of LWP delays and weakens the

SCT, contributing to a negative SW forcing (Table 5). In contrast, aerosol heating in the

PBL reduces LWP and cloud cover in the daytime (Figs. 5b and 5c) by lowering the





relative humidity in the PBL and by stabilizing the PBL (Fig. 6a), hampering the moisture supply from the surface (Fig. 6b). The reduction in cloud amount amplifies the diurnal contrast of cloud fraction and hastens the SCT, resulting in a positive SW forcing (Table 5).

The competing effects of aerosol heating in the FT versus the PBL serve to increase cloud water at night while reducing it during daytime, enhancing its diurnal cycle (Fig. 5c). Diurnally averaged, the effect of aerosol heating in the FT is dominant and leads to increased LWP and cloud cover and therefore a negative average SW forcing during the 3-day transition (Fig. 5c, Table 5). The net SW forcing is smaller than the sum

of the SW forcings via individual FT and PBL aerosol heating, indicating interactions that reduce the component forcings when combined (Table 5). Specifically, aerosol absorption in the FT slightly reduces the SW flux available for aerosol heating in the PBL, while the greater cloud breakup in the daytime reduces the reflected upwelling SW flux, in turn reducing aerosol heating in the FT. The combined effects also result in LWP and

cloud cover intermediate between the results when considered separately (Fig. 5).

In contrast to the counteracting impacts on cloud water, FT and PBL aerosol heating both inhibit entrainment by intensifying the inversion and by stratifying the PBL (Fig. 5c). The reduced PBL depth corresponds to warmer cloud tops, which emit more LW radiation upwards, leading to net negative LW forcing on days 2 and 3 despite an

increase of LWP and cloud cover (Table 5).




### 3.4. Combined effects

Comparing Tables 1, 3 and 5 it is seen that net SW forcing is weakened with all effects included because the increased LWP from aerosol heating compensates for some of the LWP loss from microphysical effects on day 2 (Table 1, Fig. 6), and the direct aerosol heating on day 1 greatly counteracts the negative radiative forcings after the aerosol layer contacts the PBL. As a result, the mean SW impact over the 3-day transition nearly vanishes (Table 1). The LW radiative forcing, however, accumulates and strengthens during the transition, and therefore is the dominant contributor to a negative average forcing during the transition (Table 1). In a nutshell, although the subsiding aerosol layer directly absorbs solar radiation and breaks up the clouds faster and more thoroughly, the CCN source serves to distribute cloud water over a greater number of drops, increasing the optical thickness of the remaining clouds but at a lower altitude, increasing both upwelling SW and LW radiative fluxes, leading to a net negative forcing. We note that day 3 net SW forcing is only negative when the aerosol is absorbing ($-1.2$ W m$^{-2}$ in Table 1); otherwise, the Twomey effect is not strong enough to counteract the reduction in cloud fraction and day 3 net SW forcing is equally positive ($1.2$ W m$^{-2}$ in Table 3).

The study of the effects of absorbing aerosol on the SCT by Y15 considered only SW forcings, which seems sensible given that studies of semi-direct effects in stratocumulus (Johnson et al., 2004) and trade cumulus (Ackerman et al., 2000; Johnson, 2005) have found SW forcings to be dominant. However, here we find interactions of aerosol and clouds in response to multiple effects leads to small net SW forcings: for example, positive SW forcing from PBL aerosol heating and microphysical effects on




dynamics offset negative SW forcing from FT aerosol heating and the Twomey effect

(Table 4). By contrast, the negative LW forcings from multiple effects (i.e., cloud water

reduction and PBL deepening) work in the same direction and result in a substantial net

LW forcing for the SCT.

Sensitivity tests with varying values of the SSA and initial number concentration

of the absorbing aerosol are summarized in Appendix A1. A decrease of SSA at 0.55-µm

wavelength from 0.88 to 0.71 hastens the SCT less but leads to a positive radiative

forcing averaged over the 3-day transition, attributable to direct absorption by the aerosol.

A decrease of the initial number concentration for the overlying aerosol with SSA of 0.88

serves to weaken its negative 3-day average radiative forcing.


**4 Variations in bulk properties of overlying aerosol layer**

**4.1.  Higher initial elevation**

Increasing the initial height of the base of aerosol layer by 400 m delays contact

with the PBL by about half a day (Fig. 7a). The delayed contact reduces the entrainment

of aerosol relative to the case with the layer starting lower, thereby hindering cloud

breakup (comparing Figs.7b-c with Figs. 2b-c). The enhanced cloud amount leads to a

much greater SW negative forcing on days 2 and 3, despite greater direct absorption

owning to the extended duration of the aerosol aloft on day 2 (Tables 1 and 6). The

delayed contact also provides for a longer duration of heating aloft and thereby a stronger

inversion on day 3 (Fig. 7e), favoring maintenance of the clouds and thus a negative SW

forcing. Despite increased LWP and cloud cover, the SCT with a higher elevated aerosol

layer is still hastened relative to the baseline (Fig. 7). The greater negative SW forcing of





the more elevated aerosol layer after its contact with the PBL ultimately leads to a more

negative 3-day mean radiative forcing to the case with the layer starting lower (Tables 1

and 6).

### 4.2.  Additional moisture

Given that observations indicate that biomass burning plumes over Namibian

stratocumulus are moister than the surrounding air (A15), next we additionally consider a

moisture perturbation relative to the baseline. As seen in Fig. 8, the moisture induces

additional SW heating and LW cooling (Figs. 8a, b), with the latter dominating. The net

cooling offsets some SW heating especially near the top of the moist layer (Fig. 8c).

Before the moist layer contacts the PBL, the additional downward LW radiative fluxes

from its moisture serve to reduce cloud-top radiative cooling and thereby drive weaker

PBL mixing that results in a more broken cloud field relative to the dry case (Fig. 9c).

Reduced LWP diminishes upwelling SW radiative fluxes, enhancing the positive SW

forcing on day 1 (Table 7). After the moist layer contacts the PBL, the entrained moist air

leads to greater LWP and cloud cover than for the baseline, despite a weaker inversion

(Figs. 8c and 9e). The increased cloud water greatly increases the net outgoing SW flux at

TOA on days 2 and 3 (Table 7), and delays the SCT relative to the dry baseline (Figs. 9b

and 9c). The SW changes in TOA radiative fluxes are seen in Table 7 to dominate the

LW changes.

When an absorbing aerosol is then added to the moist layer aloft, the SCT is faster

and more pronounced relative to the case with only a moisture perturbation (Fig. 9c).

Comparison of Tables 1 and 8 reveals that the LW forcings are comparable with and



without the additional moisture, but the SW forcings induced by indirect and semi-direct

effects are about 4 W m$^{-2}$ greater on days 2 and 3 with the moisture aloft. A thicker cloud

layer with greater cloud cover has more to lose, and the more dramatic reduction in cloud

cover during daytime predominantly changes the SW forcing. During nighttime, however,

cloud cover diminishes less as a result of the entrained moist air (Fig. 9c). The

counteracting day and night impacts on cloud cover keep the PBL depth close to that in

the absence of the additional moisture (Fig. 9d), leading to little difference in the diurnal

average LW forcing (Fig. 9f, Table 8). The net result averaged over the 3-day transition is

a modest positive SW forcing that cancels out the negative LW forcing (Table 8).


### 5. Impacts on heavily drizzling stratocumulus

The background aerosol concentrations in our simulations result in negligible

drizzle for these conditions. As SCT is often observed in association with precipitation

(e.g., Zhou et al., 2015), we next consider the impact of absorbing aerosol on the SCT of

heavily drizzling stratocumulus by reducing the $N_{a, sulfate}$ by six-fold, to 25 mg$^{-1}$.

Throughout this section the aerosol layer base is initially at 1.3 km and the layer does not

include additional moisture.

With drizzle the stratocumulus deck retains the essential features of the PBL

growth and of the thinning and dissipation of the stratocumulus layer during the SCT, but

exhibits differences associated with a much weaker diurnal cycle (Fig. 10), as also

reported by Sandu and Stevens (2011). As discussed in Sandu et al. (2008), a weaker

diurnal cycle is attributable to depletion of cloud water and stratification of the PBL via

precipitation, which limits the stratocumulus invigoration during the night. A reduced



LWP in turn lessens solar heating after sunrise, reducing daytime cloud thinning and

breakup.

As seen in Fig. 10, entrainment of aerosol inhibits drizzle and thereby thickens the stratocumulus layer. This inhibition of drizzle restores more than enough cloud water to overcome PBL drying tendencies from the increased entrainment on day 2. After sunrise, cloud cover falls sharply as the reduced drizzle strengthens the diurnal cycle. Owing to a

thicker nocturnal cloud deck and a stronger inversion from aerosol heating aloft, cloud breakup is delayed but amplified on day 2. On day 3, the aerosol heating in the presence of a stronger diurnal cycle results in a hastened SCT.

The inhibition of drizzle on day 2 allows for greater mixing and entrainment (cf. Stevens et al., 1998) despite the stronger inversion from aerosol heating aloft (Fig. 10d).

The deeper PBL is associated with cooler cloud tops that emit less LW radiation, leading to a positive LW forcing during the transition (Table 9). Such positive LW forcing is more than offset by the strong SW forcing attributable to a strong Twomey effect (relative to a cleaner baseline for this heavily drizzling case), and the net impact is therefore an amplified negative forcing (Table 9).


## 6. Discussion and conclusions

In this study we have examined the impact of an initially overlying layer of absorbing aerosol on the stratocumulus-to-cumulus transition (SCT) of lightly and heavily drizzling clouds via large-eddy simulations. Our results indicate that the

overlying aerosol can profoundly modify the breakup of stratocumulus as it advects over increasingly warm SSTs. During the transition of lightly drizzling clouds, an overlying





absorbing aerosol results in a more broken cloud field, hastening the SCT and strengthening the diurnal cycle. The hastened SCT in our simulations is primarily attributable to an increased number concentration of cloud droplets leading to faster

evaporation of more cloud water that enhances entrainment. This result holds in the presence of additional moisture in the aerosol layer and is insensitive to a 400-m increase in its initial altitude. Drizzle constitutes another degree of complexity. Its inhibition from aerosol entrainment thickens the stratocumulus and leads to a stronger diurnal cloud cycle that ultimately hastens the SCT.

The hastening of the SCT in this study is notable in contrast with Y15, who found the opposite in a similar study. The entrained aerosol in that study leads to increased cloudiness and a delay of the SCT before precipitation develops, suggesting that inhibition of precipitation is not the cause of delayed SCT in Y15. The strength of sedimentation and evaporation effects in the Y15 simulations are not obvious; we do find

a delay in the SCT for a lightly drizzling case only when sedimentation and evaporation effects are both omitted (see Appendix A2). It is noteworthy that direct numerical simulation (DNS) indicates that the sensitivity of cloud-top entrainment is substantially underpredicted in LES (de Lozar and Mellado, 2016), so in reality the microphysical effects may be considerably stronger than represented here. Another likely source of

discrepancy between our studies could be differences in model formulations. Y15 use the System for Atmospheric Modeling (SAM; Khairoutdinov and Randall, 2003) whereas here we use DHARMA (Ackerman et al., 2004). As seen in the intercomparison of de Roode et al. (2016), the evolution of cloudiness in SAM and DHARMA for that study's reference case (after Sandu and Stevens, 2011, from the observational study of Sandu et





al., 2010) is notably different in that DHARMA tends to ultimately develop a more

broken cloud field than SAM. The cloud cover in DHARMA better resembles the

satellite observations of Sandu et al. (2010) than SAM does during the SCT (Fig. 3k in de

Roode et al., 2016), but that is not necessarily proof of model skill since case study large-

scale forcings tend to be insufficiently constrained by available observations (e.g.,

Vogelmann et al. 2015). The detailed dynamical and microphysical differences between

the models warrants further investigation, and future observational studies are necessary

to provide firmer foundation of the impact of absorbing aerosol on the timing of SCT.

Our study suggests that even in the case of a hastened transition an initially

overlying absorbing aerosol layer can produce a net negative aerosol indirect and semi-

direct radiative forcings during SCT. For lightly drizzling stratocumulus, such negative

forcing is mainly attributable to greater cloud albedo from a dominant Twomey effect and

to negative LW forcing from greater cloud breakup over warmer SSTs and reduced PBL

top height from aerosol heating. Diminishing already from the interactions between

microphysical and semi-direct processes, when combined with aerosol direct SW forcing,

the net SW forcing nearly vanishes, and therefore less significant relative to the negative

LW forcing during the SCT. We recommend that such sizable LW forcings not be

neglected when considering semi-direct aerosol forcings in the context of stratocumulus

breakup. Further sensitivity tests (Appendix A1) show that when SSA at 0.5-μm

wavelength decreases further, the negative contributions can be overcome by the large

positive SW forcing via direct absorption, leading to net positive aerosol forcings. We

find it likely that similar positive forcings occur with an increase of aerosol layer

thickness.



When the aerosol layer is initially placed at a higher altitude, the extended duration of aerosol overriding the stratocumulus deck intensifies the positive SW forcing from direct absorption, while largely enhancing the negative SW indirect and semi-direct forcings from less LWP reduction owing to less entrained aerosol and a stronger inversion, leading to a more negative net forcing when averaged over the 3-day transition.

A moist layer aloft associated with outflow from a deeper continental PBL tends to intensify the radiative forcings by reducing cloud-top LW cooling and thus convective intensity and increasing the positive SW forcing before contact with the PBL, and by enhancing negative SW forcing after contact via greater LWP resulting from reduced PBL drying. The net effect of the overlying additional moisture is to modestly increase cloud water during the 3-day transition. Absorbing aerosol in the presence of additional moisture tends to break up the cloud more dramatically relative to the effect of absorbing aerosol without additional moisture aloft. The presence of moisture little affects the LW forcing but leads to substantially more net downward SW flux at TOA. Averaged over the 3-day transition, the positive SW forcing cancels out the negative LW forcing.

We note that the simulations in this study are derived from observations over the northeast Pacific Ocean (Sandu et al., 2010) whereas the characteristics of the overlying absorbing aerosol layer are based on observations from the southeast Atlantic (A15). The different large-scale meteorological conditions at these two locations may limit the generality of this study to the SCT over the Atlantic. However, we find it likely that similarly complex interactions (as summarized in Table 3) do occur. Future modeling studies based on conditions over the southeast Atlantic should be developed to evaluate the results presented here and in Y15. This study may help inform future analyses



primarily by emphasizing the complexity of competing LW and SW effects, and giving some indication of their relative strengths, which lead to a wide range of indirect plus semi-direct forcings from slightly positive to –20 W m$^{-2}$ over our 3-day simulations, depending upon assumptions made (Tables 1, 8, 9, and A1). The duration of time before

the absorbing aerosol layer makes contact with the PBL, the strength of drizzle prior to contact, the number concentration of aerosol entrained after contact and the amount of moisture accompanying the aerosol are all found to be factors of leading potential importance to regional radiative impacts of biomass burning over the southeast Atlantic and elsewhere.


   *Acknowledgments*. This research was funded by the NASA ORACLES project and the DOE ASR program. Resources supporting this work were provided by the NASA High-End Computing (HEC) Program through the NASA Advanced Supercomputing (NAS) Division at Ames Research Center. We thank Paquita Zuidema for helpful

suggestions.

APPENDIX

### a.  Sensitivity of cloudiness and aerosol radiative forcing to SSA and initial number concentration

Fig. A1 compares the 3-day transition with varying values of SSA (at 0.55-μm

wavelength) for the absorbing aerosol. As discussed earlier, the microphysical effect of aerosol acts to greatly reduce cloud water and hasten the SCT by virtue of an enhanced entrainment. This effect is also seen in the "SSA=1" case (pure scattering aerosol) in Fig. A1. The increased entrainment is reflected by the fact that the deepening of the PBL



varies little from the baseline simulation, despite substantially reduced cloud cover and

LWP. A decrease of SSA from 1 to 0.88 serves to strengthen the inversion and enhance

the diurnal cycle. These trends are greater when SSA is further reduced to 0.71, which

strengthens the inversion by ~3 K on day 2 and ~4 K on day 3, and deepens the PBL 400

m less by the end of day 3. The strengthened inversion slightly hinders cloud breakup,

while still hastening the SCT relative to the baseline (Figs. A1b and A1c). Although the

decrease of SSA amplified the net negative LW forcing via the slower deepening of the

PBL, that LW forcing is more than offset by the positive SW forcing attributable to direct

absorption by the aerosol, and therefore the 3-day mean radiative forcing increases with

the decrease of SSA. Thus, for the strongly absorbing aerosol case (SSA = 0.71) it is seen

in Table A1 that the net radiative forcing is positive on average.

560        The radiative forcing is also sensitive to the initial number concentration of the

overlying aerosol, as a five-fold reduction in $N_{a,\,absorb}$, to 1000 mg$^{-1}$, leads to the average

radiative forcing nearly vanishing during the transition (Table A1).

565        **b.  Combined effects of overlying absorbing aerosol in the absence of**

**sedimentation and evaporation effects**

As seen in Fig. A2, an overlying absorbing aerosol results in a delayed SCT when

sedimentation and evaporation effects are both omitted. The lack of microphysical

effects on dynamics isolates the influence of aerosol heating, which increases LWP

and especially cloud cover during the night and delays the SCT. We note that Y15

also found a delay in the SCT, but the similarity with this result may be coincidental.





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



FIGURES

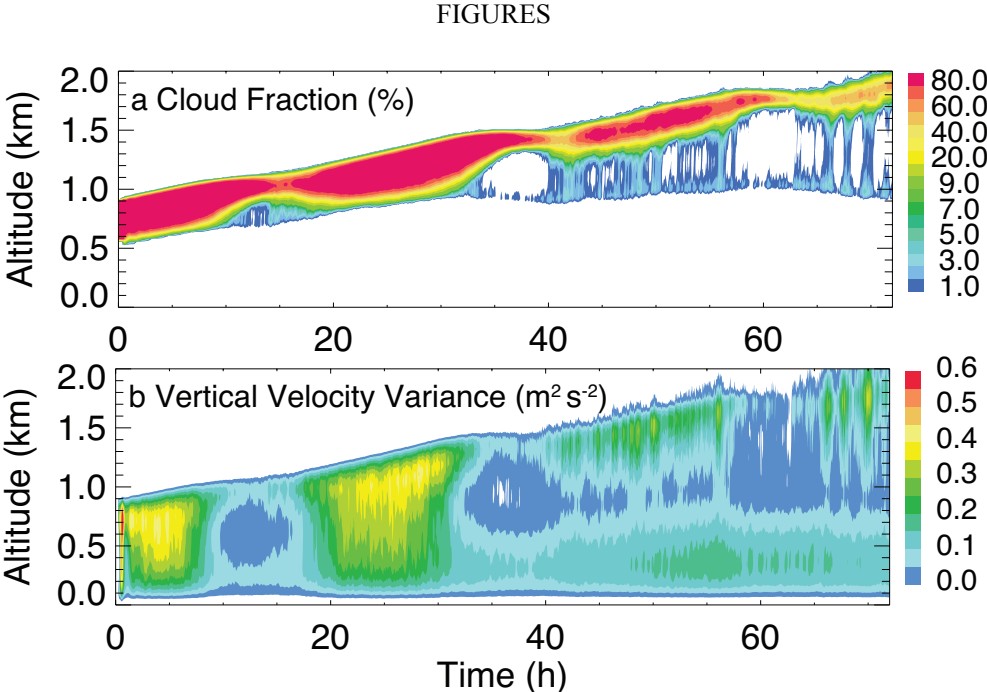

Fig. 1. Evolution of horizontal average profiles of (a) cloud fraction (where cloud water

mixing ratio exceeds 0.01 g kg$^{-1}$) and (b) vertical velocity variance for lightly drizzling

baseline case ($N_{a,\,sulfate}$=150 mg$^{-1}$).





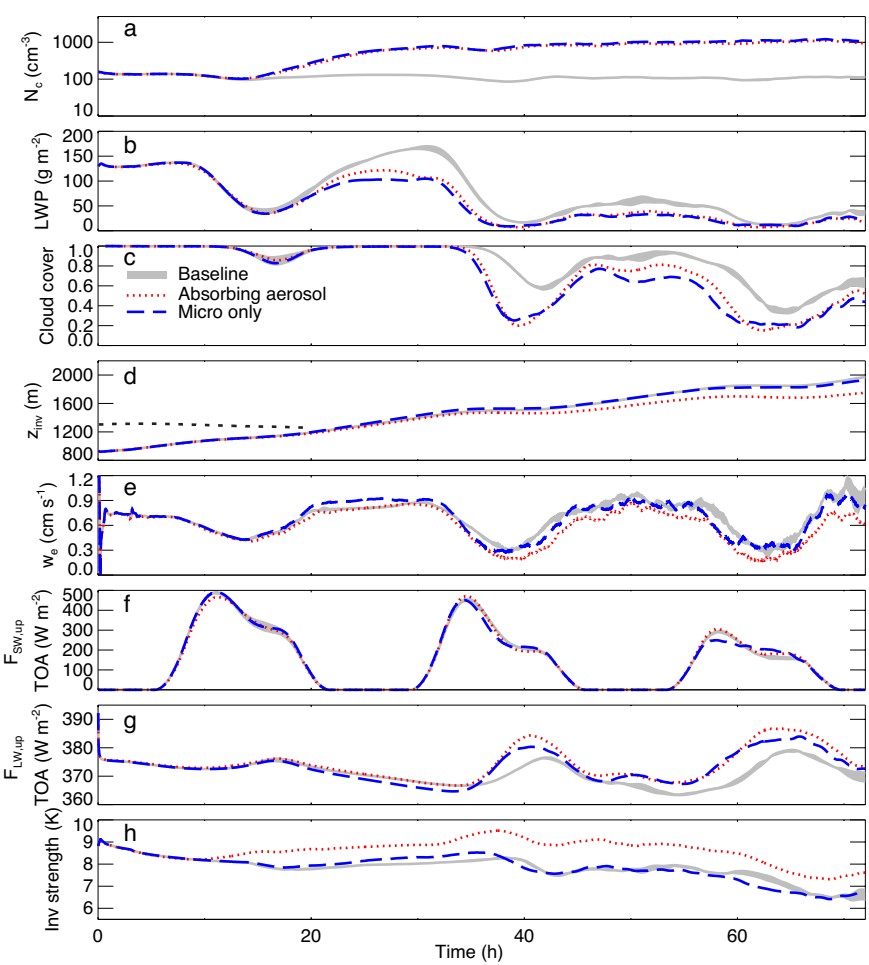

Fig. 2. Evolution of domain averages of (a) cloud droplet number concentration ($N_c$, average weighted by cloud water mixing ratio), (b) liquid water path (LWP), (c) cloud cover (columns with LWP > 10 g m$^{-2}$), (d) inversion height (height of maximum potential temperature gradient), (e) entrainment rate (difference of inversion height tendency and subsidence rate at inversion height), (f) upwelling shortwave (SW) and (g) longwave (LW) radiative fluxes at TOA and (h) inversion strength ($\Delta T$ across inversion defined as

the vertical extent with continuous positive temperature gradient). Results shown as

lagged 3-hour running averages to smooth entrainment rates. Range of 3-member lightly

drizzling baseline ensemble ($N_{a,\,sulfate}$ = 150 mg$^{-1}$) in gray. Results with absorbing aerosol

layer shown as red dotted line. Aerosol layer excluding radiative interaction shown as

blue dashed line. The black dotted line in (d) indicates the base of absorbing aerosol layer

(lowest height where $N_{a,\,absorb}$ is full strength) before contacting the boundary layer.






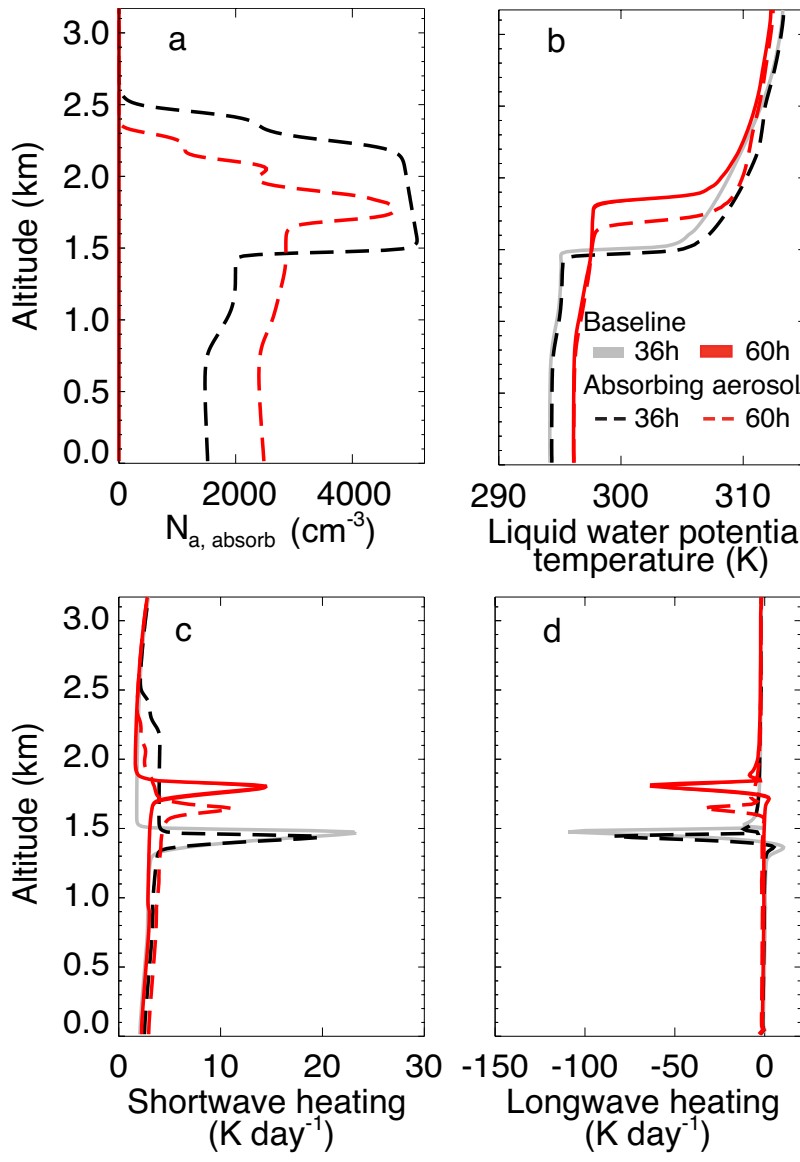

Fig. 3. Horizontally averaged profiles of (a) number concentration of absorbing aerosol,

(b) liquid water potential temperature, (c) SW heating rate and (d) LW heating rate at 36[th]

hour (gray solid line) and 60[th] hour (red solid line) for lightly drizzling baseline ensemble

($N_{a, sulfate}$ = 150 mg[-1]) and with overlying absorbing aerosol (dashed line).



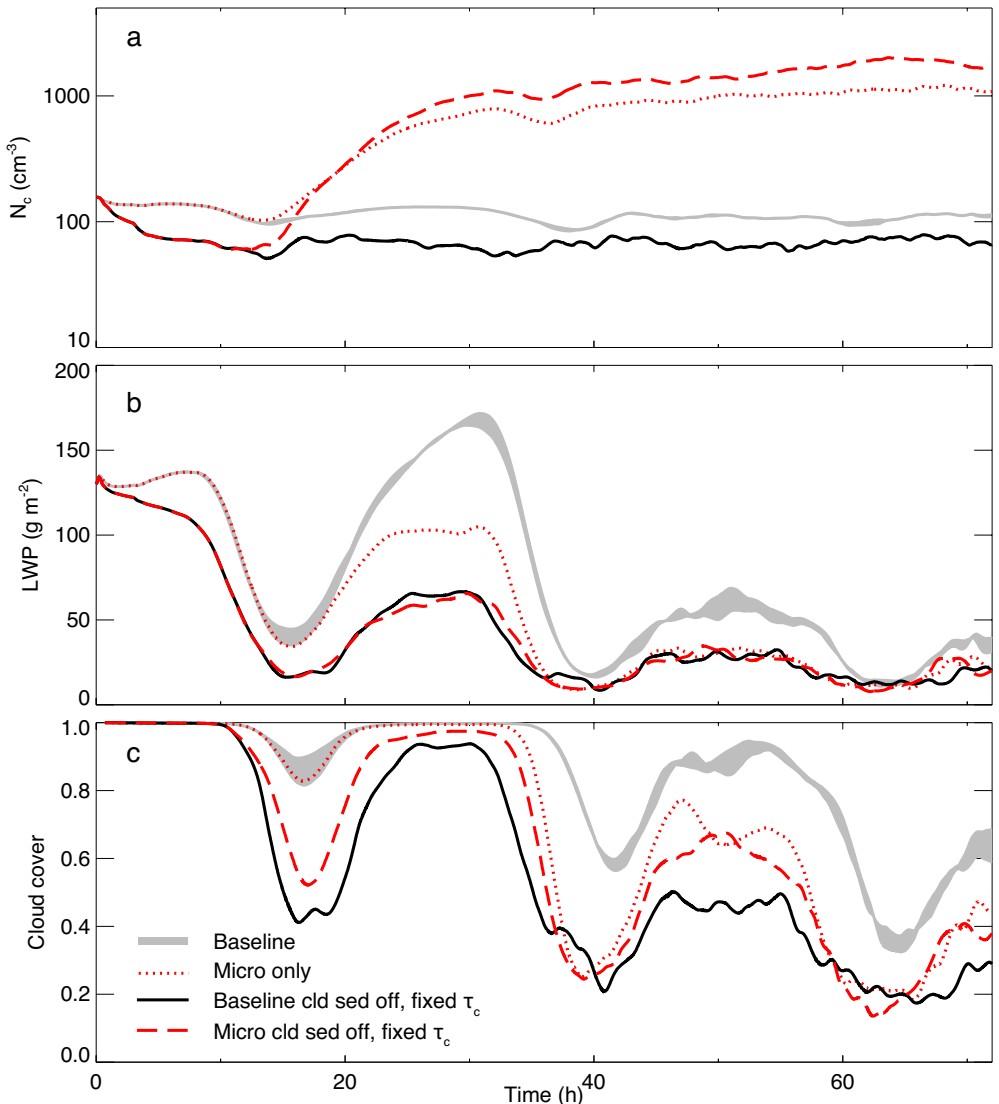

Fig. 4. As in Fig. 2 with baseline in gray and with overlying aerosol that does not affect

radiation shown with dotted red line. Baseline and overlying aerosol cases in the absence

of cloud-droplet sedimentation and with the relaxation time for diffusional growth of

cloud droplet ($\tau_c$) fixed are shown with black solid and red dashed lines respectively.





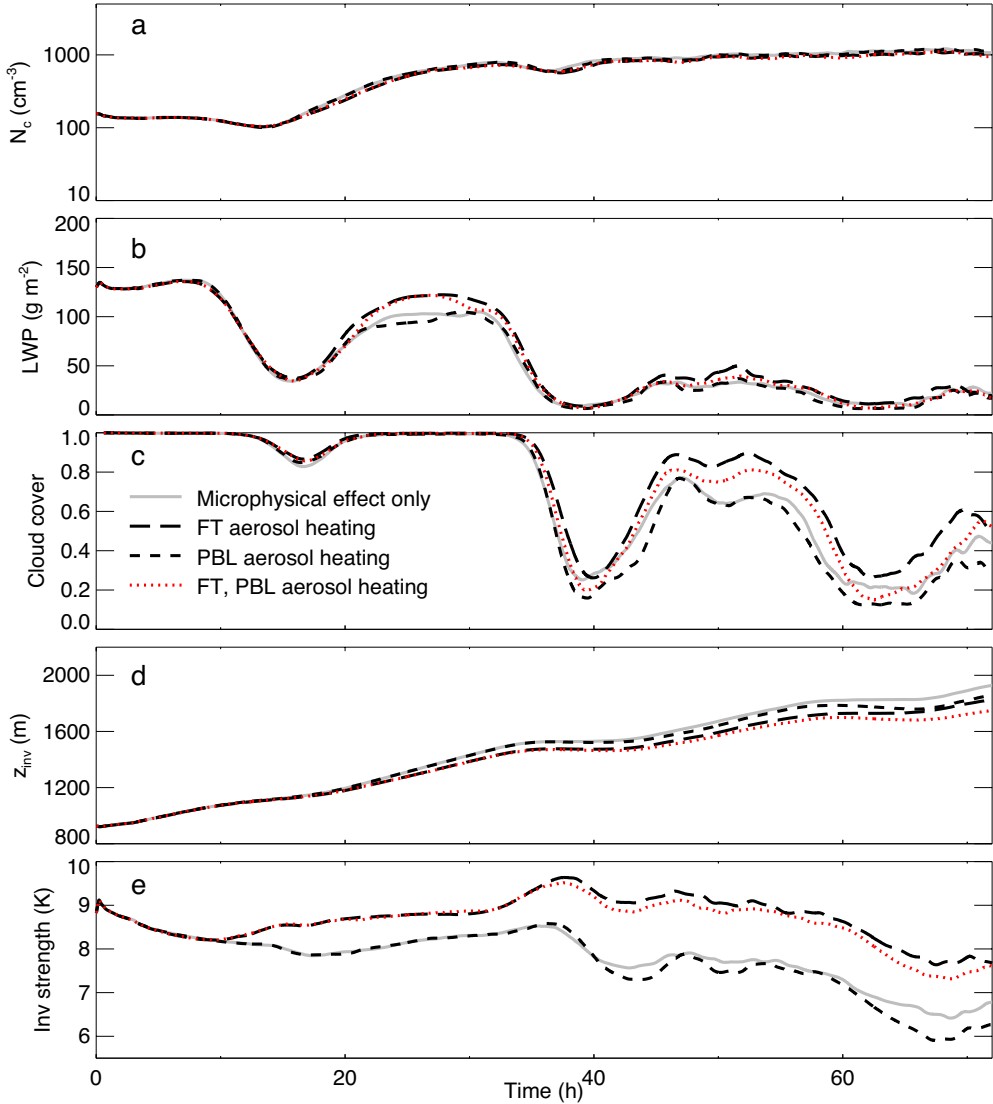

Fig. 5. As in Fig. 2. All cases include initially overlying absorbing aerosol and allow them to act as CCN. For gray solid line the aerosol does not affect radiation. For long and short dashed lines, the aerosol affects radiation only in the free troposphere (FT) and planetary boundary layer (PBL), respectively. For red dotted line there are no restrictions on aerosol affecting radiation, as in Fig. 2.





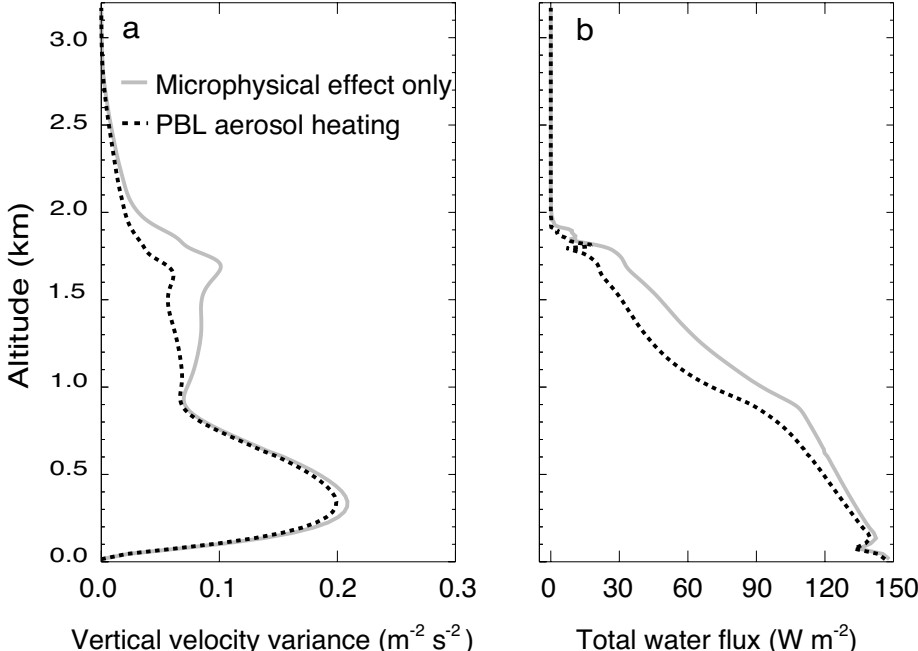

Fig. 6. Horizontally averaged profiles of (a) vertical velocity variance and (b) total water

flux averaged over 10 AM to 2 PM local time on day 3 for simulations with and without

absorbing aerosol affecting radiation in the PBL. Both simulations include microphysical

effects.







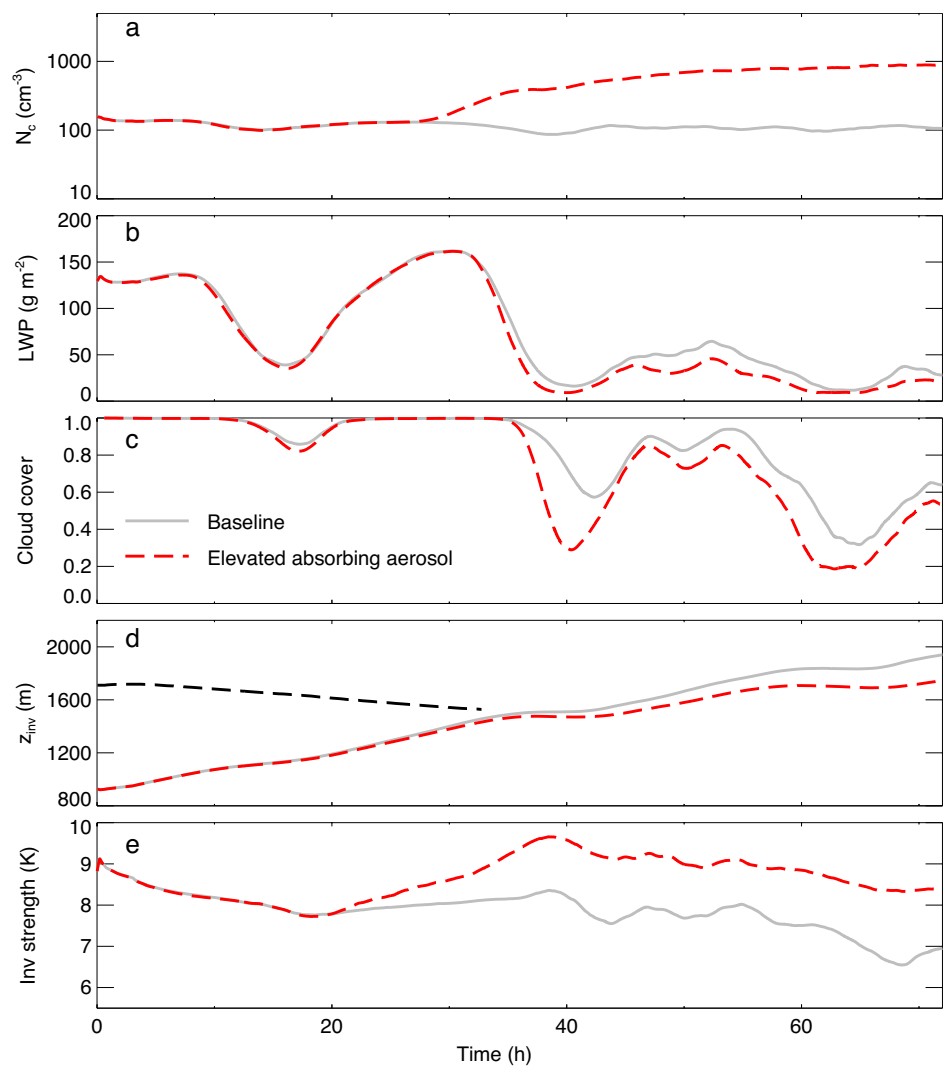

Fig. 7. As in Fig. 2. The baseline with a 3.5-km deep grid ($N_{a,\ sulfate}$=150 mg$^{-1}$) in gray.

Results with aerosol layer initially 400 m higher shown as red dashed line, with

corresponding aerosol layer base shown as black dashed line in (d).



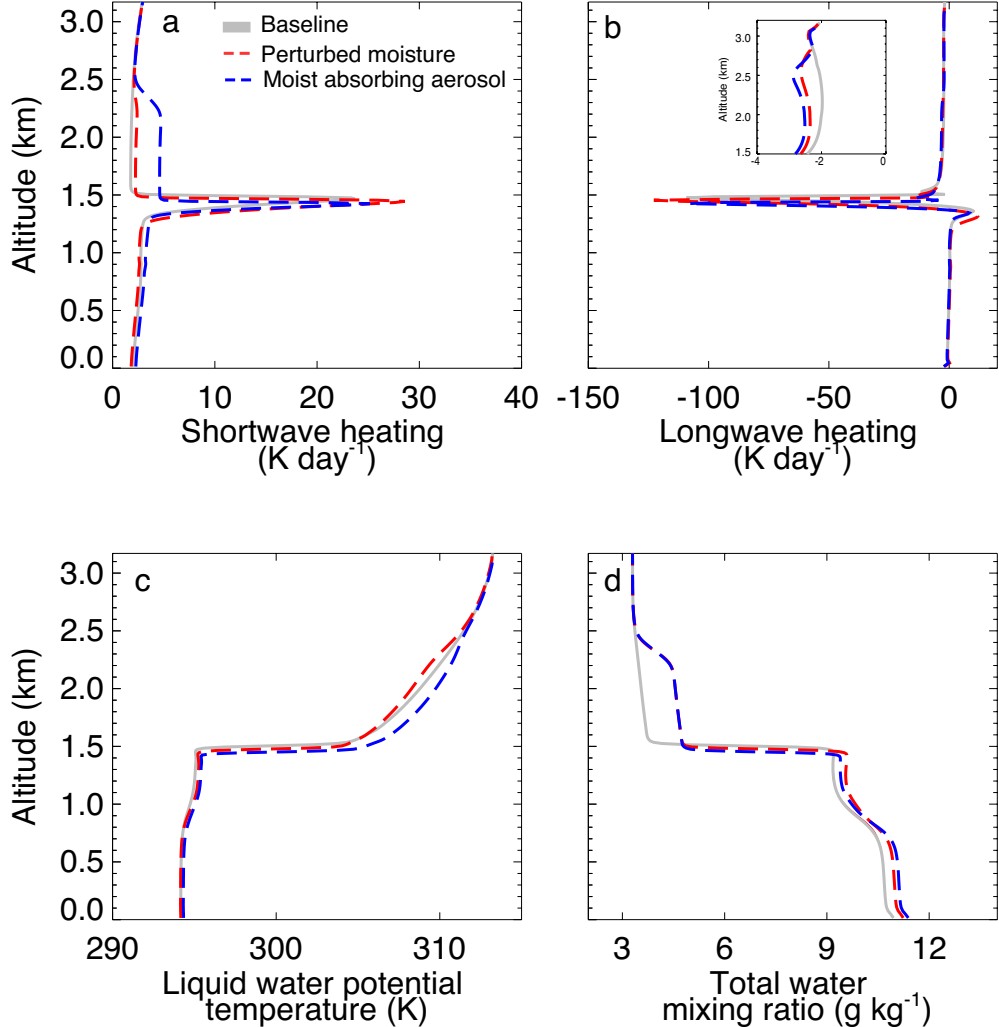


Fig. 8. Horizontally averaged profiles of (a) SW heating rate, (b) LW heating rate, (c)

liquid water potential temperature, and (d) total water mixing ratio averaged over hours

35-37 for lightly drizzling baseline ensemble ($N_{a, sulfate}$=150 mg$^{-1}$) (gray and black),

perturbed moist case (red), and perturbed moist absorbing aerosol case (blue). The sub

panel in (b) shows diurnal-average LW heating rate on day 1 from 1.5 to 3.2 km for the

above three cases.





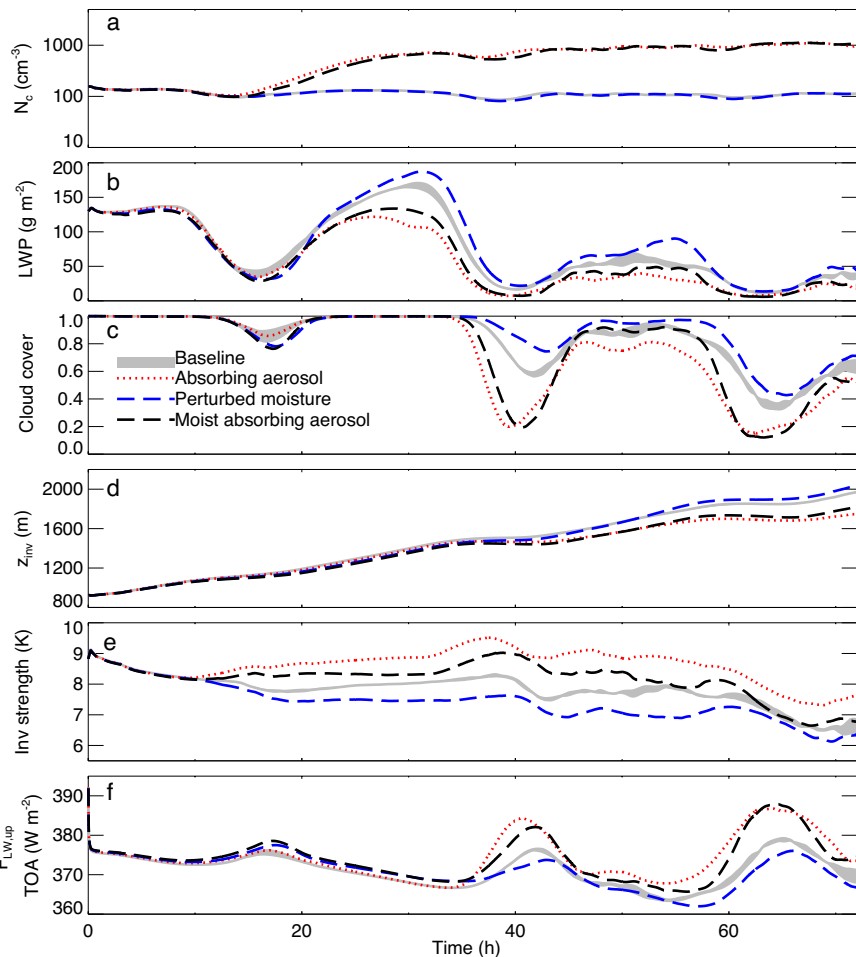

Fig. 9. As in Fig. 2. Range of three-member lightly drizzling baseline ensemble ($N_{a,}$

$_{sulfate}$=150 mg$^{-1}$) shown in gray. Results with absorbing aerosol layer shown as red dotted

line. Baseline with moist layer aloft shown with blue dashed line. Results with moist

absorbing aerosol shown as black dashed line.




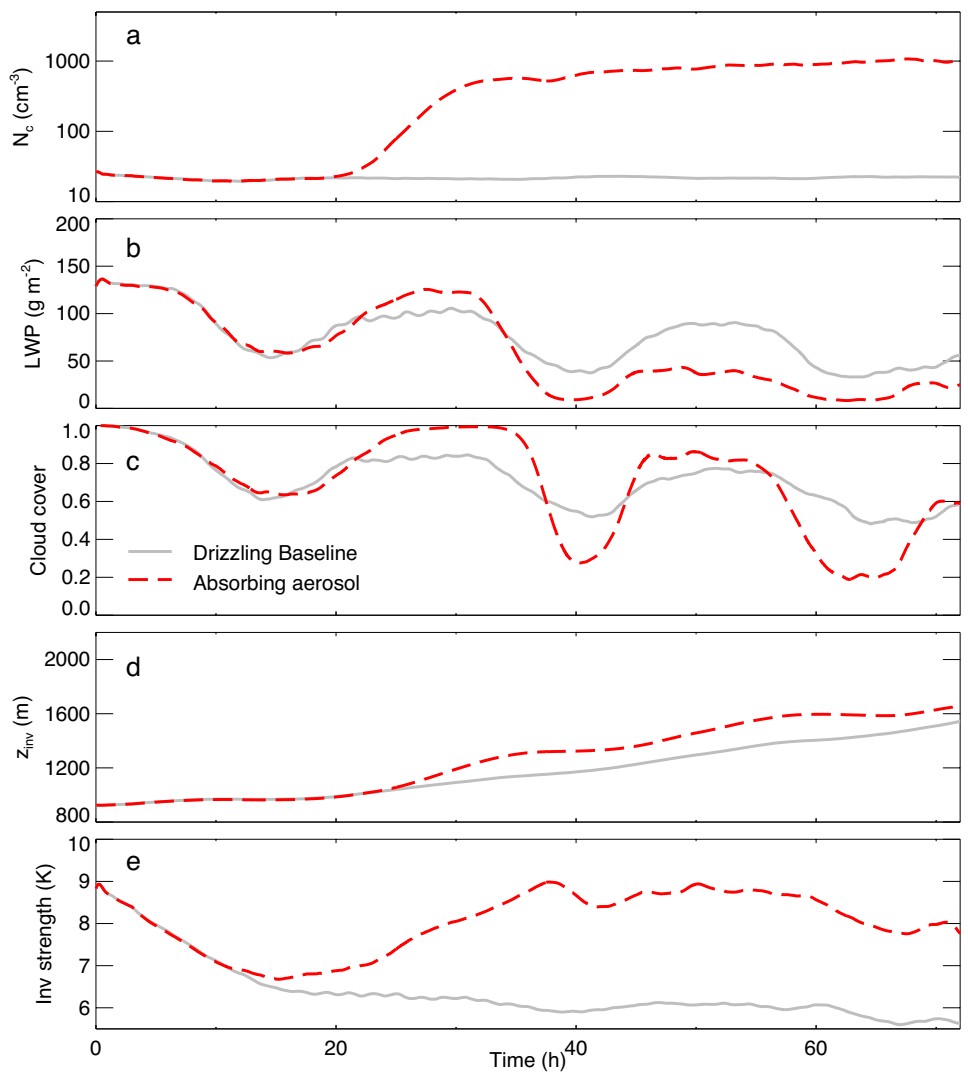

Fig. 10. As in Fig. 2 but for heavily drizzling baseline ($N_{a,\ sulfate}$=25 mg$^{-1}$ ) and with

absorbing aerosol  layer with the same $N_{a,\ sulfate}$.






Fig. A1. As in Fig. 2. Range of three-member lightly drizzling baseline ensemble ($N_{a,\ sulfate} = 150$ mg$^{-1}$) in gray. Varying single scattering albedo (SSA) of absorbing aerosol as given in legend.





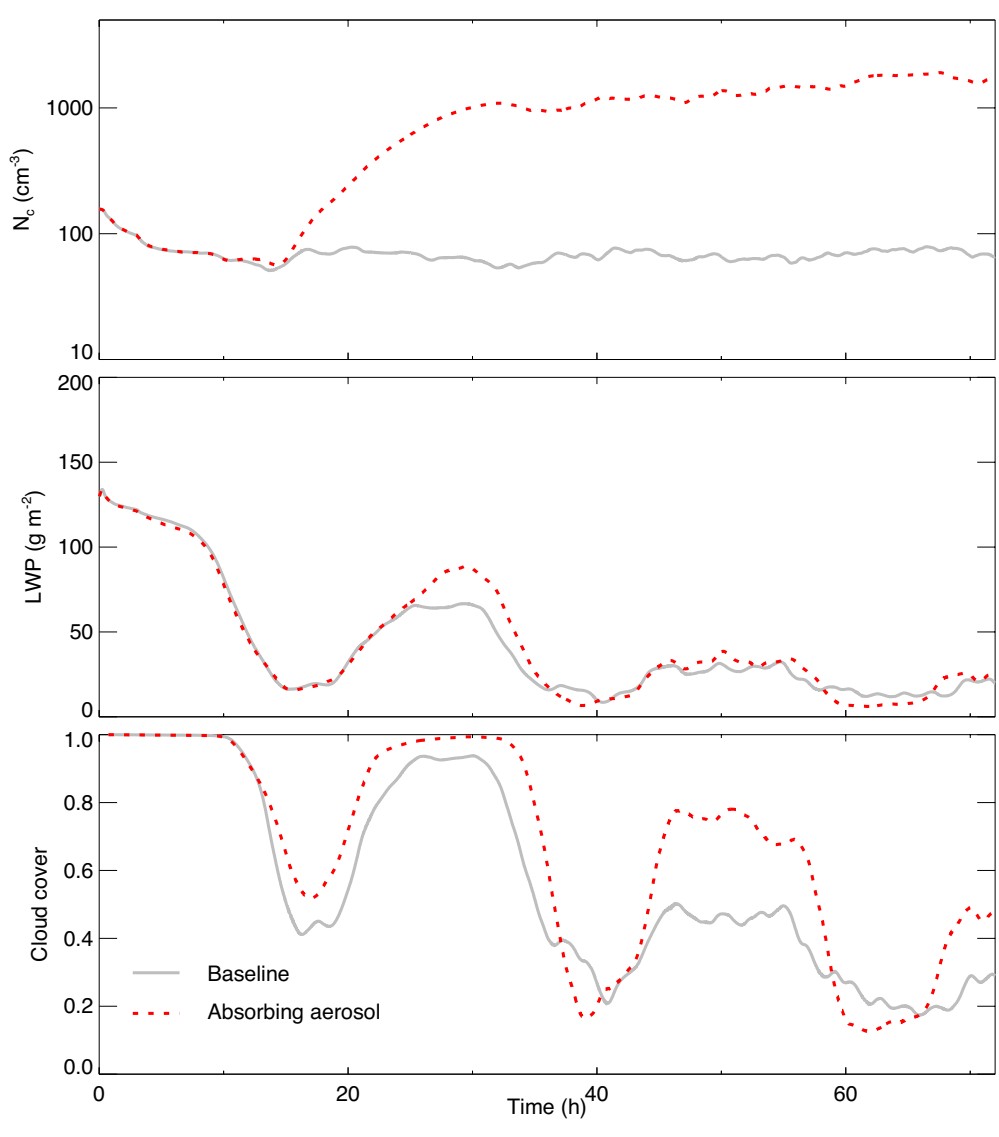


Fig. A2. As in Fig. 2 but for lightly drizzling baseline and with absorbing aerosol in the

absence of sedimentation and evaporation effects.






TABLES

Table 1. Diurnal-average direct forcing, indirect and semi-direct forcings, and all forcings (in W m$^{-2}$) from the overlying absorbing aerosol for the lightly drizzling case ($N_{a,}$ $_{sulfate}$=150 mg$^{-1}$ ) on days 1 (0-24 h), day 2 (24-48 h) and day 3 (48-72 h). The three-day average radiative forcing is indicated in the last row. Boldface indicates results exceeding the uncertainty range derived from the baseline ensemble spread.

| | Direct forcing | | | Indirect, semi-direct forcings | | | All forcings |
|---|---|---|---|---|---|---|---|
| | SW | LW | SW+LW | SW | LW | SW+LW | SW+LW |
| Day 1 | **7.3** | **-0.3** | 7.0 | -1.6 | **-0.2** | -1.8 | **5.2** |
| Day 2 | **0.8** | **-0.2** | 0.6 | -0.5 | **-2.6** | **-3.1** | **-2.5** |
| Day 3 | **-3.7** | 0.0 | **-3.7** | **-1.2** | **-6.0** | **-7.2** | **-10.9** |
| Mean | 1.5 | **-0.2** | 1.3 | -1.1 | **-2.9** | **-4.0** | **-2.7** |



Table 2. Diurnal-average changes in cloud radiative forcings ($\Delta$CRF; in W m$^{-2}$) of the overlying absorbing aerosol case relative to the lightly drizzling baseline case ($N_{a,\,sulfate}$=150 mg$^{-1}$ ). Conventions as in Table 1.


| | $\Delta$CRF TOA (W m$^{-2}$) | | |
|---|---|---|---|
| | SW | LW | SW+LW |
| Day 1 | **14.6** | -0.2 | **14.4** |
| Day 2 | **8.5** | **-2.0** | **6.5** |
| Day 3 | **2.3** | **-4.8** | **-2.5** |
| Mean | **8.4** | **-2.3** | **6.1** |







Table 3. Indirect forcing of absorbing aerosol, computed as the diurnal-average difference
in radiative fluxes at TOA (in W m$^{-2}$) of the simulation with absorbing aerosol not
affecting radiation, relative to the lightly drizzling baseline case ($N_{a,\ \mathrm{sulfate}}$=150 mg$^{-1}$ ).
Conventions as in Table 1.

|        | **Indirect forcing** | | |
|--------|------|------|-------|
|        | SW   | LW   | SW+LW |
| Day 1  | -0.7 | **0.4** | -0.3 |
| Day 2  | **2.5** | **-0.9** | **1.6** |
| Day 3  | **1.2** | **-5.2** | **-4.0** |
| Mean   | 1.0  | **-1.9** | -0.9 |





Table 4. Schematic of SW and LW radiative responses (changes in net downward fluxes

at TOA) to microphysical and thermal effects of initially overlying absorbing aerosol

layer. $N_c$ refer to cloud-droplet concentrations, CF cloud fraction, and $Z_i$ inversion height.

|  |  | SW | LW |
|---|---|---|---|
| **Microphysical effects** |  |  |  |
| **Twomey effect** | $N_c$⬆ | - |  |
| **Cloud-droplet sedimentation** ⬇ **Evaporation**⬆ | CF⬇ | + | - |
| **FT aerosol heating** |  |  |  |
| **Inversion strength**⬆ | CF⬆ | - | + |
|  | $Z_i$⬇ |  | - |
| **PBL aerosol heating** |  |  |  |
| **Aerosol heating RH decrease** | CF⬇ | + | - |
|  | $Z_i$⬇ |  | - |
| **Other** |  |  |  |
| **Warming SST** |  |  | - |





Table 5. Semi-direct forcing of absorbing aerosol, computed as the diurnal-average difference in radiative fluxes at TOA (in W m$^{-2}$) of simulations with aerosol heating restricted to the FT, PBL, or not restricted, relative to the simulation without aerosol heating. All simulations allow the absorbing aerosol to act as CCN. Boldface indicates results exceeding the uncertainty range derived from the spread of the lightly drizzling baseline ensemble.

| | | Semi-direct forcing | | |
| --- | --- | --- | --- | --- |
| | | SW | LW | SW+LW |
| FT aerosol heating | Day 1 | -1.9 | **-0.6** | -2.5 |
| | Day 2 | **-12.4** | **-0.2** | **-12.6** |
| | Day3 | **-20.6** | **2.7** | **-17.9** |
| PBL aerosol heating | Day 1 | -1.3 | 0.0 | -1.3 |
| | Day 2 | **5.5** | **-1.2** | **4.3** |
| | Day3 | **15.2** | **-3.2** | **12.0** |
| | Day 1 | -0.9 | **-0.6** | -1.5 |
| FT, PBL aerosol heating | Day2 | **-3.0** | **-1.7** | **-4.7** |
| | Day3 | **-2.4** | **-0.8** | **-3.2** |
| | Mean | **-2.1** | **-1.0** | **-3.1** |



Table 6. As in Table 1 but with absorbing aerosol layer initially located 400 m higher.

Boldface indicates results exceeding the uncertainty range derived from the spread of the

lightly drizzling baseline ensemble.

| | Direct forcing | | | Indirect, semi-direct forcings | | | All forcings |
|---|---|---|---|---|---|---|---|
| | SW | LW | SW+LW | SW | LW | SW+LW | SW+LW |
| Day 1 | **6.5** | **-0.2** | **6.3** | 4.2 | **-0.6** | 3.6 | **9.9** |
| Day 2 | **3.8** | **-0.3** | **3.5** | -11.2 | -1.9 | -13.1 | -9.6 |
| Day 3 | -3.0 | -0.1 | **-3.1** | -5.0 | -4.7 | -9.7 | -12.8 |
| Mean | **2.4** | **-0.2** | **2.2** | -4.0 | -2.4 | -6.4 | -4.2 |







Table 7. As in Table 1 but for the response of a lightly drizzling baseline to a perturbation

of moisture instead of aerosol.

|  | TOA (W m$^{-2}$) | | |
| --- | --- | --- | --- |
|  | SW | LW | SW+LW |
| Day 1 | **11.6** | **-1.3** | **10.3** |
| Day 2 | **-17.5** | **-0.2** | **-17.7** |
| Day 3 | **-9.9** | **2.4** | **-7.2** |
| Mean | **-5.2** | **0.3** | **-4.9** |






Table 8. As in Table 1 but for a lightly drizzling baseline with a moisture perturbation aloft. Boldface indicates results exceeding the uncertainty range derived from the spread of the lightly drizzling baseline ensemble.

| | Direct forcing | | | Indirect, semi-direct forcings | | | All forcings |
|---|---|---|---|---|---|---|---|
| | SW | LW | SW+LW | SW | LW | SW+LW | SW+LW |
| Day 1 | **6.1** | **-0.2** | **5.9** | -1.5 | **-0.3** | -1.8 | 4.1 |
| Day 2 | **1.8** | **-0.2** | **1.6** | **3.0** | **-2.2** | **0.8** | **2.4** |
| Day 3 | **-3.5** | 0.0 | **-3.6** | **2.8** | **-6.8** | **-4.0** | **-7.6** |
| Mean | 1.5 | -0.1 | 1.4 | 1.4 | **-3.1** | **-1.7** | -0.3 |








Table 9. As in Table 8 but for a heavily drizzling baseline ($N_{a, \text{sulfate}}$=25 mg$^{-1}$).

| | Direct forcing | | | Indirect, semi-direct forcings | | | All forcings |
|---|---|---|---|---|---|---|---|
| | SW | LW | SW+LW | SW | LW | SW+LW | SW+LW |
| Day 1 | 0.3 | -0.1 | 0.2 | -0.5 | 0.0 | -0.5 | -0.3 |
| Day 2 | **2.0** | **-0.2** | **1.8** | **-52.0** | **6.3** | **-45.7** | **-43.9** |
| Day 3 | **-3.4** | -0.0 | **-3.4** | **-9.4** | **3.4** | **-6.0** | **-9.4** |
| Mean | -0.4 | -0.1 | -0.5 | **-20.6** | **3.2** | **-17.4** | **-17.9** |








Table A1. As in Table 1 but for absorbing aerosol with different values of single scattering albedo (SSA), and only showing averages over the three-day transition. For the last case the aerosol loading is reduced five-fold.

| $N_{a,absorb}$ (mg$^{-1}$) | | Direct forcing | | | Indirect, semi-direct forcings | | | All forcings |
|---|---|---|---|---|---|---|---|---|
| | | SW | LW | SW+LW | SW | LW | SW+LW | SW+LW |
| 5000 | SSA=0.71 | **15.9** | **-0.2** | **15.7** | **-5.1** | **-5.2** | **-10.3** | **5.4** |
| | SSA=0.88 | **1.5** | **-0.2** | **1.3** | **-1.1** | **-2.9** | **-4.0** | **-2.7** |
| | SSA=1.00 | **-4.9** | **-0.1** | **-5.0** | **0.8** | **-2.5** | **-1.7** | **-6.7** |
| 1000 | SSA=0.88 | 0.2 | 0.0 | 0.2 | **2.5** | **-1.9** | 0.6 | 0.8 |