# Peer review of "Impacts of solar-absorbing aerosol layers on the transition of stratocumulus to trade cumulus clouds"

_Atmospheric Chemistry and Physics, 2017_

## Referee Comment (RC1) · P. Zuidema (Referee) · 3 Jun 2017

This manuscript examines the behavior of a stratocumulus to cumulus transition (SCT) in the presence of sunlight-absorbing aerosols distributed both inside and above the boundary layer, using the well-respected DHARMA model. The transition is based on the template of a northeast Pacific transition. Different impacts have been postulated to occur over the past 30 years in this complex regime. These are capable of either strengthening or diminishing the overall radiative impact of the low clouds on climate; this study adds to a nascent literature attempting to unravel the significance of the different effects. In this study, the increase in cloud droplet number concentra-

tion (Nc) reigns dominant in both hastening the SCT, by increasing entrainment, and in the overall radiative impact, through the Twomey effect. The study is valuable for encouraging continuing thought and discussion on the various effects and is generally well-presented.

Recommendation: Acceptance with minor revisions

Main comment:

The aerosol representation does not allow for new sources or sinks so that the total particle number concentration (Na) is conserved. From what I can tell, once the initially-specified aerosol concentrations are activated, the cloud drops also don't leave the boundary layer, in both lightly-drizzling and heavily-drizzling conditions. This would be consistent with the conservation of Na. Thus in both the sulfate and soot aerosols, the Nc approach a value of 1000/cc after 1-2 days with basically no decrease thereafter. Is this interpretation correct? There is not much discussion of the actual precipitation rates: the authors characterize light/heavy drizzle as a sulfate Na of 150 or 25/mg respectively, with no discussion of the actual precipitation rates, including of the amount reaching the surface. It would be nice to see the model precipitation rates, and to see some discussion of this feature. If it is true that Nd can't leave the boundary layer, then the conclusion that the microphysical interaction is the dominant effect is to some extent built into the model setup, it seems to me. With the power of hindsight it is easy for me to say that the post-activation Nd amount of 1000/cc is at the high end of what measured in the southeast Atlantic. The attached plot shows the number of CCN at Ascension Island, where soot is often present near the surface. At 0.4% supersaturation, an unrealistically high supersaturation, CCN only reach 1000/cc occasionally. This just meant to provide context for the modeling results.

Specific comments:

abstract, line 4: include "to cumulus" line 85: the Feingold et al 2005 study pertains to smoke-laden clouds over the Amazon. Decreases in cloudiness were explained by reductions in surface fluxes because of attenuation by the smoke layers aloft. The current study does not examine how changes in surface fluxes related to the absorbing aerosol aloft (if surface fluxes do change) affect cloudiness, and during the SCT I suspect surface fluxes most likely change because of changes in SST. It would be useful to at least provide the SST range the clouds experience during the simulations (I don't see it anywhere). But what might be more relevant to the study's focus and introduction is to mention the observational results of Wilcox et al. (2010), who found increased cloud LWP when smoke was present overhead, and Loeb and Schuster (2008) and A15, who document increased cloud cover and TOA albedo when absorbing aerosols are present aloft. These observational results seem to suggest support for a negative (cooling) semi-direct effect (though in truth given how much the thermodynamic profiles in the aerosol composites shown in A15 fig. 14 differ from those depicted in the study in review, one has to wonder if perhaps associated changes in the large-scale circulation end up dominating the cloud response). in line 116 and in other places (line 202, the authors connect humidity increases with outflow from a deep continental boundary layer. It's also worth mentioning the role of the large-scale circulation, as for much of the year the smoke flows westward rather than eastward. Strong easterly winds aloft are needed to advect both the aerosol and moisture offshore, with some portion caught up in an anticyclonic circulation induced by a heat low over southern Africa, that further disperses both aerosol and humidity offshore. This characterization is the focus of Adebiyi and Zuidema, 2016. lines 197-206: a table of the different experiments would be useful, including within it a column listing the figures in which their results are shown. line 204: should 'impact' be preceded by 'microphysical'? line 238: worth mentioning that higher-level clouds are not considered. line 243 or elsewhere: it would be useful to see the precipitation rates and vertical structure associated with both the lightly and heavily drizzling cases. . .and the SST values imposed on the simulation. Figs 1, 2 and elsewhere: It would also be useful to mark the daylight (e.g. 6am-6pm LT) portions on the figures, and include mention of the starting time of the simulation in the caption of at least fig. 1. I also don't see discussion anywhere of how the large-scale subsidence

is prescribed. It is not connected to the radiative warming I'm pretty sure, which would also be good to mention. section 3.3: it looks to me from fig. 5 that the microphysical effect is still included from the absorbing aerosol experiments intended to focus on the semi-direct effect, is that correct? section 4.1, line 384: I don't think the simulations allow the radiative heating to translate into anomalous ascent. ERA-I reanalysis (A15, fig. 15 and the simulations of Sakeada et al 2011 do suggest the larger-scale subsidence is weaker when absorbing aerosols are present). It's worth mentioning. line 383: 'owning' should be 'owing'

Figures: see comment 7 above Tables: I had difficulty interpreting Table 4, perhaps it was just my printout. The physical processes sometimes span two lines, other times not. Why does increased evaporation not get a '+' in the SW column and '-' in the LW column? Why are other SW/LW columns left blank?

Tables 7 and 8: I think this is the first time I see an ensemble of the same simulations mentioned. would be useful to mention in section 2 somewhere if ensembles were indeed done.

References:

Adebiyi, A.and P. Zuidema, 2016: The role of the southern African easterly jet in modifying the southeast Atlantic aerosol and cloud environments. Q. J. R. Meteorol. Soc., 142, p. 1574-1589 doi: [10.1002/qj.2765] Loeb, N. G., and G. L. Schuster, 2008: An observational study of the relationship between cloud, aerosol and meteorology in broken low-level cloud conditions. J. Geophys. Res., 113, D14214, doi:10.1029/2007JD009763. Wilcox, E., 2010: Stratocumulus cloud thickening beneath layers of absorbing smoke aerosol. Atmos. Chem. Phys., 10, 11 769– 11 777, doi:10.5194/acp-10-11769-2010.

-Paquita Zuidema

2017.

**May 18 - November 30, 2016 black carbon mass conc.**

SP2

single
particle
soot
photometer

Art Sedlacek

Aug 13

June 24

Aug 30

**corrected aerosol absorption coefficients (PSAP)**

PSAP

Virkkula-Bond
|differences|

Mobile
Aerosol
Observing
System
Team

**cloud condensation nuclei concentration @ 0.4 supersaturation**

CCN

Janek Uin

June 24

Aug 13

Aug 30

**Fig. 1.** data from Ascension Island, located at the end of the SCT. note the bottom panel in particular)

---

## Referee Comment (RC2) · Anonymous Referee #2 · 6 Jul 2017

General Comments:

This study performs a comprehensive investigation of the impact of solar-absorbing aerosol and moisture on the Stratocumulus-to-Cumulus Transition of lightly and heavily drizzling clouds. By using large-eddy simulation, it is indicated that the overlying aerosol can substantially modify the stratocumulus due to an increase in the number concentration of cloud droplets induced by entrained aerosol. Meanwhile, the impacts of additional moisture in aerosol layer are also investigated. The results are generally well presented and structured, and the topic is suitable for publication in Atmos. Chem. Phys. after addressing some specific comments listed below.

[Figure]

Specific Comments:

In the baseline and further simulations, ammonium sulphate are assumed to be uniformly distributed vertically. Since it is a typical anthropogenic aerosol and mainly formed near the surface, its concentration is more likely to decrease with height through the troposphere. Thus, it would be better to characterize its vertical distribution according to climatological profile that provided by pre-existing long-term simulation using chemical transport model or available observations.

Several parallel numerical simulations are conducted to isolate the microphysical effect, semi-direct effect and direct effect of aerosols. Using an additional table in Sect. 2 to illustrate the numerical experiment design and how these aforementioned effects are derived based on these simulations may help clarify the link and difference.

Another issue is that the input of meteorological conditions and the characteristics of aerosol layer are derived from different locations, northeast Pacific Ocean and southeast Atlantic, respectively. Using the observations in the same region could make this work more practical and representative.

Technical Corrections:

Page 8 Line 155: Some basic information like initial time and spin-up duration need to be specified here. It would help to understand the following figures since the x axis are relaxation time.

The starting time of the simulation was already stated (see line 143). We now reiterate in the caption of Fig. 1 that "The simulation starts at midnight local time".

We have added "After ~2 h of boundary layer turbulence spin-up (Fig. 1b)" at lines 251-252.

Fig.1 and 2: It would be more clear to label the local time in the x axis.

Line 199: It is better to use "model top" rather than "domain".

---

## Author Comment (AC1) · 17 Aug 2017

**P. Zuidema (Reviewer #1)**

This manuscript examines the behavior of a stratocumulus to cumulus transition (SCT) in the presence of sunlight-absorbing aerosols distributed both inside and above the boundary layer, using the well-respected DHARMA model. The transition is based on the template of a northeast Pacific transition. Different impacts have been postulated to occur over the past 30 years in this complex regime. These are capable of either strengthening or diminishing the overall radiative impact of the low clouds on climate; this study adds to a nascent literature attempting to unravel the significance of the different effects. In this study, the increase in cloud droplet number concentration (Nc) reigns dominant in both hastening the SCT, by increasing entrainment, and in the overall radiative impact, through the Twomey effect. The study is valuable for encouraging continuing thought and discussion on the various effects and is generally well-presented.

Recommendation: Acceptance with minor revisions

Main comment:

The aerosol representation does not allow for new sources or sinks so that the total particle number concentration (Na) is conserved. From what I can tell, once the initially-specified aerosol concentrations are activated, the cloud drops also don't leave the boundary layer, in both lightly-drizzling and heavily-drizzling conditions. This would be consistent with the conservation of Na. Thus in both the sulfate and soot aerosols, the Nc approach a value of 1000/cc after 1-2 days with basically no decrease thereafter. Is this interpretation correct? There is not much discussion of the actual precipitation rates: the authors characterize light/heavy drizzle as a sulfate Na of 150 or 25/mg respectively, with no discussion of the actual precipitation rates, including of the amount reaching the surface. It would be nice to see the model precipitation rates, and to see some discussion of this feature. If it is true that Nd can't leave the boundary layer, then the conclusion that the microphysical interaction is the dominant effect is to some extent built into the model setup, it seems to me. With the power of hindsight it is easy for me to say that the post-activation Nd amount of 1000/cc is at the high end of what measured in the southeast Atlantic. The attached plot shows the number of CCN at Ascension Island, where soot is often present near the surface. At 0.4% supersaturation, an unrealistically high supersaturation, CCN only reach 1000/cc occasionally. This just meant to provide context for the modeling results.

While cloud droplet number concentration is prognostic in our simulations (clarification added at line 164), the reviewer's interpretation is correct in that there is no aerosol consumption via collision-coalescence in these simulations (contrast to Y15 now noted at lines 499-503. The properties for the absorbing aerosol layer are based on published studies as cited, with the number concentrations for the sake of heating rates and extinction, as stated. Coincidentally, the absorbing aerosol number concentrations are

comparable to those in Y15. In future work that is ongoing we will use vetted measurements from the ORACLES field campaign to explore sensitivities beyond those considered in this study, among them aerosol consumption and absorbing aerosol number concentrations (see line 503).

We now show cloud-base precipitation rates in Figs. 2 and 10 and refer to them at lines 258 and 440.

Specific comments:

1. abstract, line 4: include "to cumulus"

Added.

2. line 85: the Feingold et al 2005 study pertains to smoke-laden clouds over the Amazon. Decreases in cloudiness were explained by reductions in surface fluxes because of attenuation by the smoke layers aloft. The current study does not examine how changes in surface fluxes related to the absorbing aerosol aloft (if surface fluxes do change) affect cloudiness, and during the SCT I suspect surface fluxes most likely change because of changes in SST. It would be useful to at least provide the SST range the clouds experience during the simulations (I don't see it anywhere).

We only model the atmosphere and over rather short time spans here, and thus do not consider any effects on ocean temperatures. We have added text clarifying our approach on lines 144-146: "Surface fluxes are computed following similarity theory as in Ackerman et al. (1995). Note that because sea surface temperature is prescribed, it is not impacted by changes in the overlying atmosphere." The SSTs used for the SCT setup are documented by Sandu and Stevens (2011) as well as by de Roode et al. (2016) and in the intercomparison specifications; we now also document them on lines 133-135: "Following Sandu and Stevens (2011) and de Roode et al. (2016), SST increases steadily from 293.75 K at 0 h to 299.17 K at 72 h...".

But what might be more relevant to the study's focus and introduction is to mention the observational results of Wilcox et al. (2010), who found increased cloud LWP when smoke was present overhead, and Loeb and Schuster (2008) and A15, who document increased cloud cover and TOA albedo when absorbing aerosols are present aloft. These observational results seem to suggest support for a negative (cooling) semi-direct effect (though in truth given how much the thermodynamic profiles in the aerosol composites shown in A15 fig. 14 differ from those depicted in the study in review, one has to wonder if perhaps associated changes in the large-scale circulation end up dominating the cloud response).

Our simulations capture a variety of responses when variations of the height of the absorbing aerosol layer and properties of the ambient atmosphere are considered, but feedbacks with large-scale dynamics are beyond the scope of this small-scale, atmosphere-only modeling study. We have added references to Wilcox et al. (2010), Loeb and Schuster (2008) and A15 to the introduction (lines 9-20 where we summarize previous studies. Global modeling studies are now recommended at line 542.

3. in line 116 and in other places (line 202), the authors connect humidity increases with outflow from a deep continental boundary layer. It's also worth mentioning the role of the large-scale circulation, as for much of the year the smoke flows westward rather than eastward. Strong easterly winds aloft are needed to advect both the aerosol and moisture offshore, with some portion caught up in an anticyclonic circulation induced by a heat low over southern Africa, that further disperses both aerosol and humidity offshore. This characterization is the focus of Adebiyi and Zuidema, 2016.

We now specifically mention the easterly component of equatorward flow when first discussing the SCT in the Introduction, mention that the humidity aloft "accompanies the absorbing aerosol that results from biomass burning" on lines 80, 97 and 105, and state the following on lines 104-107: "We note that in our modeling framework it is simply assumed that the model domain is advected equatorward by the trade winds, thus implicitly treating the flow aloft as being easterly, despite observations that indicate circulation in the South Atlantic to be far more complex (e.g., Adebiyi and Zuidema, 2016)."

4. lines 197-206: a table of the different experiments would be useful, including within it a column listing the figures in which their results are shown.

We have added such a table in Section 2.

5. line 204: should 'impact' be preceded by 'microphysical'?

No—here we investigated the total impact (direct, semi-direct, and indirect effect) of overlying absorbing aerosol on heavily precipitating stratocumulus, not just the microphysical impact.

6. line 238: worth mentioning that higher-level clouds are not considered.

Added on line 24.

7. line 243 or elsewhere: it would be useful to see the precipitation rates and vertical structure associated with both the lightly and heavily drizzling cases. . .and the SST values imposed on the simulation.

We have included the precipitation rates at the cloud base to Fig. 2. The SST values are addressed in our response to comment #2.

8. Figs 1, 2 and elsewhere: It would also be useful to mark the daylight (e.g. 6am-6pm LT) portions on the figures, and include mention of the starting time of the simulation in the caption of at least fig. 1. I also don't see discussion anywhere of how the large-scale subsidence is prescribed. It is not connected to the radiative warming I'm pretty sure, which would also be good to mention.

We now indicate the nominal night time (6 pm – 6 am LT) in gray shading in Fig. 1a-and reiterate the simulation starting time in its caption. The treatment of large-scale subsidence in the SCT setup is documented by Sandu and Stevens (2011) and de Roode

et al. (2011), and we now also provide that information on lines 133-137 "Following Sandu and Stevens (2011) and de Roode et al. (2016) ... a uniform divergence of large-scale horizontal winds of $1.86 \times 10^{-6}$ s$^{-1}$ is imposed up to an altitude of 2000 m, above which the large-scale subsidence is constant."

9. section 3.3: it looks to me from fig. 5 that the microphysical effect is still included from the absorbing aerosol experiments intended to focus on the semi-direct effect, is that correct?

Correct. We have clarified our approach by adding the following text on lines 324-326: "By doing so we build upon the results of the previous section, effectively evaluating semi-direct effects in the presence of microphysical effects, rather than in their absence."

10. section 4.1, line 384: I don't think the simulations allow the radiative heating to translate into anomalous ascent. ERA-I reanalysis (A15, fig. 15 and the simulations of Sakeada et al 2011 do suggest the larger-scale subsidence is weaker when absorbing aerosols are present). It's worth mentioning.

Large-scale subsidence in our simulations is indeed prescribed, and beyond the additional detail added in response to comment #8, we have also added the following text on lines 137-139: "Because the large-scale subsidence is imposed rather than interactive, we omit any possible decrease in subsidence associated with solar heating by absorbing aerosol (cf. Sakaeda et al. 2011)."

11. line 383: 'owning' should be 'owing'

Corrected.

12. Figures: see comment 7 above

Please see our response to that comment.

13. Tables: I had difficulty interpreting Table 4, perhaps it was just my printout. The physical processes sometimes span two lines, other times not. Why does increased evaporation not get a '+' in the SW column and '-' in the LW column? Why are other SW/LW columns left blank?

We have improved the readability of that table (now Table 5) by adding a comma to the cell that includes two effects, replacing blanks with zeros, and adding clarification to the caption: "Plus signs refer to positive responses, negative signs to negative responses, and zeros to negligible or absent responses".

14. Tables 7 and 8: I think this is the first time I see an ensemble of the same simulations mentioned. would be useful to mention in section 2 somewhere if ensembles were indeed done.

It was mentioned on line 177 that the baseline case is an ensemble of three simulations. We now clarify that Fig. 1 shows a single baseline ensemble member whereas Fig. 2

shows the baseline ensemble range (lines 252 and 258.

**Reviewer #2**

General Comments:

This study performs a comprehensive investigation of the impact of solar-absorbing aerosol and moisture on the Stratocumulus-to-Cumulus Transition of lightly and heavily drizzling clouds. By using large-eddy simulation, it is indicated that the overlying aerosol can substantially modify the stratocumulus due to an increase in the number concentration of cloud droplets induced by entrained aerosol. Meanwhile, the impacts of additional moisture in aerosol layer are also investigated. The results are generally well presented and structured, and the topic is suitable for publication in Atmos. Chem. Phys. after addressing some specific comments listed below.

Specific Comments:

In the baseline and further simulations, ammonium sulphate are assumed to be uniformly distributed vertically. Since it is a typical anthropogenic aerosol and mainly formed near the surface, its concentration is more likely to decrease with height through the troposphere. Thus, it would be better to characterize its vertical distribution according to climatological profile that provided by pre-existing long-term simulation using chemical transport model or available observations.

Agreed. Please see our response to the main comment of the first reviewer.

Several parallel numerical simulations are conducted to isolate the microphysical effect, semi-direct effect and direct effect of aerosols. Using an additional table in Sect. 2 to illustrate the numerical experiment design and how these aforementioned effects are derived based on these simulations may help clarify the link and difference.

Agreed. Please see our response to comment #4 of the first reviewer.

Another issue is that the input of meteorological conditions and the characteristics of aerosol layer are derived from different locations, northeast Pacific Ocean and south-east Atlantic, respectively. Using the observations in the same region could make this work more practical and representative.

Agreed. We already noted and addressed this head-on on lines 536-552 Use of meteorological and aerosol conditions over the Atlantic is the subject of a future study that we have begun, using very recently released measurements. The present study is intended to identify the most relevant aspects of observed variability for our next study, as summarized in the concluding sentence.

Technical Corrections:

Page 8 Line 155: Some basic information like initial time and spin-up duration need to be specified here. It would help to understand the following figures since the x axis are

relaxation time.

The starting time of the simulation was already stated (see line 143). We now reiterate in the caption of Fig. 1 that "The simulation starts at midnight local time".

We have added "After ~2 h of boundary layer turbulence spin-up (Fig. 1b)" at lines 251-252.

Fig.1 and 2: It would be more clear to label the local time in the x axis.

Please see our response to comment #8 of the first reviewer.

Line 199: It is better to use "model top" rather than "domain".

Done.